# The Design of Health Promoting Outdoor Environments for People with Young-Onset Dementia—A Study from a Rehabilitation Garden

**DOI:** 10.3390/ijerph21081047

**Published:** 2024-08-09

**Authors:** Nina Oher, Jonna Tingberg, Anna Bengtsson

**Affiliations:** 1Department of People and Society, Faculty of Landscape Architecture, Horticulture and Crop Production Sciences, Swedish University of Agricultural Sciences (SLU), Alnarp, P.O. Box 190, 234 22 Lomma, Sweden; anna.bengtsson@slu.se; 2Memory Clinic, Skåne University Hospital, 205 02 Malmö, Sweden; jonna.tingberg@skane.se

**Keywords:** young-onset dementia (YOD), evidence-based design (EBD), outdoor environment, garden, supportive environment, dementia-friendly environment, health promoting, people living with dementia (PLwD), nature

## Abstract

Health-promoting outdoor environments designed for people living with dementia (PLwD) has proved to be an effective non-pharmacological intervention for treatment of symptoms and improved well-being. However, for individuals with Young-Onset Dementia (YOD), who have particular symptoms and needs, the content and design of these environments are underexplored. This study aimed to explore the needs of individuals with YOD in a garden setting, to generate design-related knowledge for ‘dementia-friendly’ outdoor environments, while contributing to the field of Evidence-Based Design (EBD). An 8-week long nature-based program was carried out in Alnarp’s rehabilitation garden, a specifically developed garden based on research from e.g., landscape architecture, environmental psychology and medical science. The study used a triangulation of qualitative methods including six participants with YOD and a multidisciplinary team of five staff members. Content analysis was used for all gathered data, including 17 semi-structured interviews with participants with YOD and with staff. Data collection and analysis was performed based on the evidence-based Quality Evaluation Tool (QET). The study led to a target group adapted version of the QET for people with YOD containing 20 developed environmental qualities for designers to pursue in therapeutic gardens, including the additional quality of *Calmness.* A progression was noted, as a result of perceived positive effects during the intervention, indicating possible change and development of the group’s needs and preferences in the outdoors.

## 1. Introduction

### 1.1. Dementia

Dementia is considered one of our major public diseases [1] and is recognised by the World Health Organization (WHO) as a public health priority [2]. It is estimated globally that more than 55 million people have dementia today, with nearly 10 million new cases each year [2]. This rapid increase is a result of the world’s aging population due to advances in the medical field, which have generated an increase in life expectancy [3,4]. Dementia is a collective term for several degenerative diseases that affect the brain with consequences on memory, cognitive abilities, and the ability to perform daily activities [2]. It has physical and psychological, as well as social and economic consequences for people living with dementia (PLwD), as well as their careers, families and society at large [2]. Although dementia is not an inevitable or natural part of biological aging, age is the strongest known risk factor [2,5]. However, dementia can also affect younger people [6,7]. Young-Onset Dementia (YOD) is defined as the onset of symptoms before the age of 65, but where some are afflicted as early as in their 30s [8,9]. This group accounts for up to 9% of all dementia cases [2]. It is estimated that 1 in 1000 people under the age of 65 develop dementia [10], which corresponds to around 3.9 million people living with YOD in the world today [11].

In Sweden, there are currently around 160,000 PLwD [12], and the number is expected to double by the year 2050 [13]. This means major challenges and increased demands on healthcare in the coming years [13]. In addition, it is estimated that up to 10,000 individuals are living with YOD [14], with the majority being between 60 and 65 years of age [15].

### 1.2. Young-Onset Dementia

YOD differs from Late Onset Dementia (LOD) in that younger individuals are more likely to have rare forms of dementia, which affect behaviour and social functioning. They also experience significantly higher psychological and physical distress as a result of their illness [16]. People with YOD are often forced to stop working, which can have negative economic outcomes and lead to a significant loss of social networks [17,18]. Experiencing significant stigma and discrimination is also common, due to frequent misconceptions about YOD regarding age and symptoms [16,18], as dementia is not a disease one would expect to find in younger people [19]. This may in turn lead to social isolation [17,18]. Boredom and lack of meaningful employment is another challenge often faced by younger PLwD [20], as they often find themselves “… *locked away in their own homes for safety*” [9]. This loss of physical and social simulation can furthermore lead to great confusion, anxiety, and a more rapid progression of symptoms [9].

The most common disease causing dementia in people under 65 is the same as for LOD, namely Alzheimer’s disease, followed by vascular dementia, frontotemporal dementia [11], alcohol-related dementia and, accounting for a smaller proportion, dementia with Lewy bodies [21]. Nevertheless, YOD-related symptoms tend to be more neuropsychiatric rather than cognitive [17,22] as challenges with speech and language, behavioural or personality changes tend to be more of a problem than “traditional” symptoms of memory loss [23,24]. Symptoms may furthermore include difficulties with planning or carrying out every-day tasks and decision-making, as well as problems with visual perception, assessment of depth, distance and colour [24].

Although people under 65 are used to being in control [25], are physically strong and have higher energy levels than older people [17], they are often offered post-diagnostic support aimed at older people with memory problems [24,25,26], where too little or no consideration is given to their interests, needs for physical activity [23,24,25] and other specific needs of support [17,27]. Additionally, research found that people with YOD experienced the disease’s impact on their thoughts and memory as a painful loss of personality and self-esteem [28]. Therefore, to increase the well-being of people with YOD, there is a need to provide care aimed at reducing their memory loss, as well as supporting autonomy and control [9]. But in order to provide this kind of age-appropriate care, we need to increase knowledge about what must be in place for people living with YOD [29].

### 1.3. Non-Pharmacological Intervention

Although there are some medications that can help manage dementia symptoms, pharmacological research has not yet led to a cure for Alzheimer’s disease or other types of dementia [2]. However, WHO [2] believes that much can be done to support PLwD. Being physically active, as well as participating in activities and social interactions that stimulate the brain and maintain daily functions are some examples. There is thus a need to explore non-pharmacological interventions [30] to find treatment options that can benefit PLwD and contribute to improved well-being, quality of life [31,32], reduced symptoms and slower cognitive decline [30].

More and more research indicate that the outdoor environment, its content, design [33] and activities carried out there, can have a positive impact on patients’ health, well-being and quality of life [34,35,36,37]. The outdoors therefore has the potential to become a valuable resource in healthcare contexts [38]. Clatworthy et al. [39] state that there is a significant body of research showing that garden-based interventions can benefit people with mental health problems.

Several systematic reviews on the benefits of activities in the outdoor environment for PLwD have been conducted, focusing on, for example, horticultural therapy [40,41], therapeutic gardens [37] or garden-based activities in community settings [30].

Out of all the studies included in the above-mentioned reviews [30,37,40,41] only one study focused on people with YOD. That study was done by Hewitt et al. [42] who conducted a structured gardening program with YOD participants, which showed positive effects linked to self-identity, purposeful activity and mood, as well as autonomy and sociability. The authors suggested that a meaningful guided activity program may maintain or improve well-being in the presence of cognitive decline. They also pointed to the importance of people with YOD being part of a group in a non-hospital environment with peers, and showed increased sociability, self-confidence and a sense of belonging for the participants as a result.

Although there is some literature on the positive health benefits of outdoor activities for individuals with YOD, albeit very limited, there is still a great need to understand what these outdoor environments should look like and contain to best suit the specific target group. Environmental design for dementia has been shown to be an effective non-pharmacological intervention for treating symptoms [43], justifying the need and development of ‘dementia-friendly’ outdoor environments.

### 1.4. Dementia-Friendly Environments

The physical environment has the potential to facilitate or hinder PLwD in their daily lives. A well-designed environment, based on the experiences of PLwD [44], can help them maintain their abilities, increase independence and provide meaningful engagement [45]. Dementia friendly design has been described as a useful approach for creating environments that are friendly, inclusive and supportive [46]. A dementia-friendly environment is defined as an environment that promotes independence and supports well-being, has familiar surroundings, provides easy access and wayfinding, supports meaningful tasks, supports participation in daily activities, promotes safety, security and comfort [45].

A dementia-friendly design is furthermore important as environments that are not designed with the needs of PLwD in mind may be experienced as confusing and disorienting, at worst disabling and even dangerous by the target group [46]. A variety of cognitive, physical and sensory impairments can be related to dementia [43], with potential consequences for design of the physical environment. Examples are issues with perception, which is common for PLwD. For example, high-contrast floors or any decisive separation of one area from another, as well as dark patterns, can often be perceived as a change in depth [47]. Altered experiences and responses to temperatures is also experienced by PLwD [48], due to changes in the brain that affect the body’s ability to regulate its temperature [49]. Additionally, PLwD do not always remember to dress appropriately for colder weather and may not even realise the fact that they themselves are freezing [50]. Sensory impairments related to hearing, sight, smell and (to a lesser extent) taste have also been documented [51]. For PLwD, sensory experiences become the primary means of understanding the world, as cognitive function declines [51]. However, this occurs with a reduced ability to integrate the experiences to understand the context, which makes PLwD very sensitive to sensory stimulation. Furthermore, it is common for people with YOD in healthcare settings to experience problems in relation to both sensory overstimulation and sensory deprivation [9]

Previous studies on dementia-friendly design of outdoor environments in nursing homes for PLwD (65 years and older) and dementia-friendly neighborhoods for older PLwD have been conducted [36,52,53]. They highlight important qualities such as sensory stimulation, meaningful activities, socialisation, accessibility, orientation, safety, comprehensibility, comfort and familiarity. It is further known that reminiscence, sensory stimulation and memory stimulation can provide physical and emotional comfort [54,55] as well as reduce anxiety for PLwD [56]. Additionally, it has been pointed out that routes with many starting points and repetitive elements [57] as well as excessive emphasis on safety features [58] should be avoided.

Research focusing on people with YOD has also been conducted, but instead with a focus on the indoor environment. Qualities of importance are mentioned, such as wayfinding, easy to understand, safety, security, social interaction with peers, meaningful activates, memory stimulation, outdoor access [59], sensory stimulation, physical activity, memorable and familiar sense of place, stimulating architecture, integration of nature and community [9] are mentioned.

For people with YOD, who struggle with both sensory overstimulation and sensory deprivation, contact with nature, physical exercise, social interaction and multisensory elements is said to be helpful in balancing the level of stimulation [9]. However, studies describing dementia-friendly qualities related specifically to people living with YOD and their needs in the outdoor environment seems to be lacking.

### 1.5. Evidence-Based Design

Evidence-based design (EBD) has the potential to provide practitioners with the knowledge needed to develop dementia-friendly environments. EBD is an internationally established field of research, which has emerged in connection with an increased awareness of the importance of the physical environment as part of a good healing environment [60]. EBD is also an approach, that is, a way of working, which has gained significant popularity in healthcare architecture and aims to improve patient and staff well-being, accelerate the healing process, relieve stress and increase safety [61]. The Center for Health Design [62] describes EBD as “*the process of basing decisions about the built environment on credible research to achieve the best possible patient, staff and operational outcomes*”.

In order for the research evidence to be used correctly, it is important that its relevance is assessed against the conditions of the current project, i.e., the context of the project, such as patient group, project type and environment for which the design is intended [63,64,65]. In addition to supporting the design process itself, evidence-based environmental qualities can support the evaluation of environmental design [43], which is a crucial part of the EBD concept [66,67]. Evaluations highlight the importance of continuously building and developing evidence within EBD in order to expand the knowledge base [68].

### 1.6. The Quality Evaluation Tool

While EBD is a working method and an approach to design (as well as a field of research), the Quality Evaluation Tool (QET) is an evidence-based design tool that helps practitioners analyse outdoor environments, in relation to specific target groups and activities [33]. The aim is to support evidence-based design processes for outdoor environments in healthcare contexts [69]. QET hence serves the EBD process by concretely highlighting what the specific target group needs in terms of the outdoor environment. QET thus contributes to the part of EBD that deals with acquiring knowledge and understanding of the target group.

The QET is based on theories of restorative and supportive environments, mainly from the field of environmental psychology, combined with evidence of specific environmental qualities and design concepts [70]. Underlying theories of specific importance are the Attention Restoration Theory (ART) [71,72], the Stress Reduction Theory [73] and the Supportive Environment Theory (SET) [74]. These theories are based on the premise that humans, trough evolution, are adapted to a life in nature. They furthermore focus on restorative and supportive environments, including those relating to recovery from stress and stress-related symptoms such as impaired concentration. People risk exhausting their resources in trying to meet the demands of every-day life, which can affect their abilities, health and well-being. Environments that support people’s recovery processes and promote the renewal of these depleted resources are called restorative environments [70,75]. Supportive environments are defined as functionally effective environments, consisting of patient-centered or psychologically supportive characteristic that can help patients manage stress and facilitate recovery [76,77]. Supportive environments are perceived as comprehensive, accessible, safe and meaningful [66].

A current issue is that academics experience difficulties in effectively reaching out to design practice, while designers have difficulty implementing academic production in practice. This is described as the “research-practice gap” [78]. QET aims to reduce this gap by presenting 19 environmental qualities significant in the outdoors, based on relevant theory and extensive research findings, for practical use in projects and design processes (Appendix A).

The tool furthermore includes a model that highlights different zones of contact with the outdoors, as well as a suggested 4-phase working model [33].

#### 1.6.1. Evidence-Based Environmental Qualities for the Outdoors

The 19 environmental qualities are divided into two main groups. The first group (A) identifies and describes six environment qualities (A1–A6) to enhance the comfort of people in outdoor environments. It prevents risk factors by creating a comfortable, safe and secure outdoor environment. These qualities are a prerequisite for a positive outdoor stay and should therefore be present in the entire environment. The second group (B) describes thirteen environmental qualities (B1–B13) concerning access to nature and surrounding life, which promote stimulation of the senses and mind, and correspond to qualities that people desire and prefer in the outdoors. They encourage people to go outdoors, either to find social or physical activities, or to find solitude and peacefulness [33]. The B-qualities offer a spectrum of supportive and health promoting environments. This spectrum, called the Sensory Opportunity Spectrum (SOS), thus offers environmental qualities with both restorative and stimulating potential, suited for different needs and, for example, stages of a rehabilitation process [74]. The environmental qualities are placed on a gradient within this spectrum, from most stimulating (B1) to most restorative (B13). The 19 environmental qualities must be seen in relation to specific needs, wishes, preferences and activities linked to the target group for which the design is intended. Together, they encompass a wide range of needs and preferences, consistent with people in general as well as people with special needs and conditions in health care settings [33,69].

The Triangle of Supportive Environments (Figure 1) is a model linked to the Supportive Environment Theory, and aims to explain how certain environments, more than others, can help support people who feel unwell [74]. The model presents a gradient: from garden and nature environments that are dominated by more peaceful natural environments important for recovery (i.e., inwards involvement), to environments that are dominated by social activities (i.e., outwards involvement). For target groups that are sensitive to overstimulation, the need for more peaceful environments and restoration is the most important. This is illustrated by the wide base of the triangle, where the most extensive and initial needs are located. As the individuals start to feel better, they might move up the triangle and seek out other environmental qualities of a more social and stimulating nature. However, for target groups sensitive to understimulation, the triangle is turned upside down (Figure 2) so that the wider base is instead placed by the social and stimulating qualities, higher up on the gradient, since these correspond to what the target group needs or desires most [33,79]. For further description of the QET, see Bengtsson et al., 2024 [33].

#### 1.6.2. Model Highlighting Different Zones of Contact with the Outdoors

Additionally, the environmental qualities of QET are investigated and/or considered in different zones, in accordance with the principle model Four Zones of Contact with the Outdoors. This model identifies four zones in which health-promoting interaction with the outdoor environment can take place: Zone (1) From inside the building, e.g., through a window, Zone (2) In transition zones, e.g., conservatory, greenhouse, balcony, Zone (3) In the nearby/immediate environment, such as a garden, and Zone (4) In the surrounding environment, i.e., beyond the healthcare building and its associated immediate environment [33,79]. The model aims to assist those involved in the design and planning processes by highlighting the importance of an examination of the environment, all the way from inside the building to the outdoors [33,79].

### 1.7. Aim

Many studies have shown positive effects of being in outdoor environments for PLwD, such as improved well-being and quality of life. However, the specific needs of individuals with YOD in these settings remain understudied. It is crucial to identify the environmental qualities important to people with YOD, to facilitate the design of dementia-friendly outdoor environments tailored for this particular target group. The study’s hypothesis is that QET’s evidence-based environmental qualities is relevant to individuals with YOD. The aim is therefore also to test QET and to target group adapt the tool, if needed, to better suit people living with YOD.

In conclusion, this study aims to explore the needs of individuals with YOD in a garden setting, to generate design-related knowledge for ‘dementia-friendly’ outdoor environments, while developing QET and contributing to the field of EBD.

## 2. Materials and Methods

This study was carried out as a collaboration between the Memory Clinic in Malmö, Sweden, and researchers from the Department of People and Society at the Swedish University of Agricultural Sciences (SLU). The collaboration included a larger project where a new roof garden at the Memory Clinic was to be built. In relation to this larger project, the current study aimed to gather knowledge about dementia-friendly outdoor environments that could then be considered in relation to the context of the new roof garden. In addition, the study aimed to capture and describe the special needs of people with YOD, a group relevant in relation to the planned use of the Memory Clinic’s roof garden.

A qualitative research methodology was used, suitable for exploring and gaining a deep understanding of a particular problem by collecting the participants’ experiences, perceptions and behaviours in relation to the problem at hand [80]. The study involved a content analysis of seventeen semi-structured interviews with study participants (individuals with YOD and staff), workshop notes, as well as diary entries and logbooks (from staff) to achieve richness in data.

In order to gain more detailed knowledge in relation to the physical environment and highlight environmental qualities of importance to people living with YOD, the QET tool [69] was used in the present study. It was employed as a basis for data collection during interviews, as a guide during the data analysis and as a structure for the result presentation. Research has pointed out a need for models, tools and structured approaches for the implementation of research-based knowledge in design processes [81,82,83], as well as the need to identify specific environmental qualities of the outdoors that promote health and well-being [84]. This summarises what QET is intended for [79] and further justifies the use of the tool, which in relation to the purpose of the study was considered to be both relevant and useful.

QET is furthermore based on a review that summarises a large body of research on design and content of garden environments in healthcare settings [69]. By using QET, the current study develops and adds to the existing knowledge base. The evaluation and development of the QET is in line with the authors’ intention for the initial version of the tool, which is to generate “*future versions of the QET focusing on different health conditions/diseases*” [69].

### 2.1. Study Design

At the time of the study, a Memory Health unit at the Memory Clinic carried out in-depth examinations, assessments and interventions for PLwD. The unit consisted of occupational therapists, physiotherapists, counsellors and psychologists who worked together at the Memory Clinic in several multidisciplinary teams. Their common goal was to support the patient’s ability for activity and participation, based on the needs of the individual patient. The ambition was to extend this supportive work to the outdoor environment once the Memory Clinic’s new roof garden was completed.

To learn about the needs and desires of people with YOD, an intervention research study was designed. Qualitative research methods, as used in this study, are well suited for intervention research that takes place in the field [85]. In the health sciences, qualitative research methods are used to study the nature of various phenomena, by addressing questions such as “*what works for whom, when, how, and why*” [86,87]. Garden interventions open up for the possibility of getting answers to a large number of questions as they offer more than just contact with nature. They typically provide opportunities for people to interact with others, enable them to engage in meaningful activity, develop specific knowledge and skills, and involve physical activity [40]. Research-in-context can, in addition to better understanding the phenomenon being studied, also result in a close fit between research and practice [88]. Randomisation or control groups are not typical methods in qualitative research [87] and were thus not used in this study.

### 2.2. Setting

Because an outdoor environment did not yet exist at the Memory Clinic, the study was instead carried out in Alnarp’s rehabilitation garden.

Alnarp’s rehabilitation garden is a specially designed garden in southern Sweden, where studies on the importance of natural environments for health and well-being have been conducted since 2002. The two-hectare garden is located on the campus of the Swedish University of Agricultural Sciences and belongs to the university’s research infrastructure [74]. Theories on restorative [71,72,74,76,77] and supportive outdoor environments [74], as well as knowledge from disciplines such as landscape architecture, environmental psychology, horticulture, physical therapy, occupational therapy, and medical science underpin the garden’s design [74].

In this article, the rehabilitation garden will simply be referred to as “*the garden*”. The results of the study derive above all from the intervention situation, i.e., the participants’ experiences during their visit to the garden.

### 2.3. Participants

#### 2.3.1. Recruitment

The study participants included individuals with YOD from the Memory Clinic, as well as staff from both the Memory Clinic and from Alnarp’s Rehabilitation Garden. Individuals who visit the Memory Clinic come to be examined and treated for a suspected cognitive disorder (dementia). It may be due to difficulties with remembering or in relation to orientation, language disorders or personality changes.

The individuals are often referred to the Memory Clinic by a general practitioner at a local health clinic [89]. Primarily it is PLwD in the early stages of their illness, healthy enough to make the journey, who visit the Memory Clinic. PLwD in the later stages of their illness are cared for in their own homes or in nursing homes. All forms of dementia are progressive, which means that the symptoms may be relatively mild at first but worsen over time. Dementia is said to develop in three stages—early, middle and late. These are sometimes called mild, moderate and severe cognitive decline, as it describes how much the symptoms affect a person [90].

Selection and recruitment of PLwD for the study was carried out by nurses at the Memory Clinic. These PLwD participated in regular follow-ups at the clinic and were therefore well known to the recruiting nurses. Only those PLwD who were deemed capable of giving informed consent, i.e., competent to make decisions, were approached.

The ambition was to be able to recruit PLwD with a variety of dementia diagnoses, as well as to carry out a strategic selection in terms of age and gender. What took precedence, however, was to find PLwD who wanted to participate in the study, who could manage to travel to and from the garden by taxi and have the energy to participate in the garden program twice a week. The recruiting staff also prioritised PLwD who lived outside the city and did not already avail of some form of organised daytime activity. Additional inclusion criteria include the ability to use the toilet independently, no dysphasia/aphasia, to understand and speak Swedish and no simultaneous participation in other health science intervention studies.

Recruitment took place within the clinic’s “younger patients” group, consisting of individuals who developed dementia before the age of 65. This group met the study’s inclusion criteria. They also constituted an important group in relation to the planned use of the Memory Clinic’s new roof garden, as previously mentioned.

#### 2.3.2. Individuals with YOD from the Memory Clinic

Six individuals with YOD participated in the study, four women and two men. All had been diagnosed with either Alzheimer’s disease or vascular dementia in the past one to two years and were assessed to suffer from mild to moderate dementia at the time of the study (Table 1). As patients at the Memory Clinic, they had been examined and undergone a clinical dementia investigation and meet the clinical criteria for cognitive disorder (dementia) according to DSM V and ICD 10. Since they had developed dementia before the age of 65, they belonged to the YOD category. All individuals were still residing at home. A seventh YOD participant was scheduled to partake in the study (male born in 1951, diagnosed with Lewy body dementia two years earlier, assessed to have mild dementia) but was unable to participate due to planned holiday abroad.

#### 2.3.3. Staff Included in the Garden Study

The study involved two Memory Clinic staff members, both experienced nurses, who had worked extensively with patients suffering from cognitive disorders such as dementia. The nurses were previously known to the YOD participants in the study. Additionally, the study involved three members of the Rehabilitation Garden team: an occupational therapist, a horticultural engineer, and a physiotherapist and ergonomist. This team had several years of experience conducting intervention research studies and working with various patient groups in the garden, including individuals with stress-related mental illness (e.g., depression and burnout syndrome) [91], stroke [92], Parkinson’s disease [93] and with migrants from war-torn areas to facilitate integration and workplace induction [94].

#### 2.3.4. The Garden Program

The program in Alnarp’s Rehabilitation Garden was designed to immerse the individuals with YOD in a supportive outdoor environment where they could choose activities to participate in, based on their ability and interest, while being accompanied by staff to guide and support them. At each visit to the garden, one of the nurses and two members from the rehabilitation garden were present. The program spanned eight consecutive weeks with two half-days (3.5 h) per week in the garden, resulting in 16 sessions. Every visit followed the same schedule, starting with tea and an overview of the garden activities on offer. This was followed by various voluntary activities in the garden. Each visit ended with a light lunch together, where the YOD participants also reflected on experiences during the day. The garden activities included walks, planting, harvesting, picking flowers, feeding fish in the pond, picking and drying apples, replanting flowers, tying wreaths, preparing meals and more. Active participation, as well as more passive forms of participation (e.g., sitting, resting, and observing) were possible. The staff used an open and flexible approach that also welcomed individual requests and suggestions from the YOD participants.

### 2.4. Data Collection

This study investigated the preferences and needs of individuals with YOD, including their perceptions, experiences, likes, dislikes and wishes in relation to the outdoor environment. The data collection was carried out in collaboration between the first and last author.

#### 2.4.1. Data from YOD Participants

Semi-structured qualitative interviews were conducted with the YOD participants. The semi-structured approach enables interviews to stay focused while at the same time giving the interviewer autonomy to explore relevant information that may appear during the interview, which can further increase the understanding of the phenomenon being explored [95]. The YOD participants were interviewed individually twice during the course of the intervention, once in the middle (after about 4 weeks) and once towards the end (after almost 8 weeks), to reduce the length of each session and improve the possibility for the YOD participants to have enough energy to answer all questions. The ambition was further to capture both their early impressions and experiences in the garden, as well as their impressions and experiences after a longer time spent in the garden. Considering the YOD individuals’ existing memory problems, it was important to conduct the interviews on two separate occasions. This approach aimed to minimise the possibility of them forgetting impressions, feelings and thoughts by the time of the interview. The interviews were held in the rehabilitation garden, in the main building to offer privacy and minimise distraction, in a room with windows overlooking the garden. The participant to be interviewed was brought into the room while the rest of the group continued with activities in the garden. Both researchers participated in the sessions. The interviews were audio recorded and later transcribed.

#### 2.4.2. Data from Participating Staff

In order to obtain rich and comprehensive data, and to include another perspective, information was also collected from the staff with a focus on the YOD participants’ experiences, use and needs in the outdoor environment. This included written information, individual interviews and a workshop. During each session in the garden, staff were asked to observe the whole group as well as each individual with YOD. These observations were made specifically for the purpose of this project, not as part of any treatment and without access to medical records. The staff’s observations and reflections were documented in logbooks and diaries at the end of each visit, to capture the daily events. Staff also took notes during the reflection sessions held in the garden at the end of each day, in an attempt to capture the YOD participants’ thoughts and experiences. Questions were asked such as “*What has been good/less good today?”* and *“What was the most important/greatest experience for you today?*” The logbooks, diaries, and notes were handed over to the researchers (i.e., the first author) for analysis at the conclusion of the study. Semi-structured interviews were conducted individually with each staff member once, shortly after the end of the intervention. Two out of the five staff interviews were conducted with only one researcher, for practical reasons. The interviews, conducted at the rehabilitation garden, were audio recorded and later transcribed. Finally, a workshop was conducted at the same location with the staff (from both the rehabilitation garden and the Memory clinic) about 6 weeks after the study for a joint discussion, so as not to miss relevant information or reflections that may have emerged in the time after. Both researchers were present during the workshop.

#### 2.4.3. Interviews

All interviews were based on the QET. The tool was used for the opportunity to help and inform an EBD process, by concretely highlighting what this target group needed in relation to the outdoor environment. This was done primarily by using the tool’s evidence-based environmental qualities as a basis for constructing the interview guides.

The first interview with the YOD participants had a more general approach. It mainly focused on aspects related to (A) Being Comfortable in the outdoors, i.e., on what makes it possible to feel comfortable enough in the outdoor environment to both be able and dare to use it. Questions were asked such as “*Which parts of the garden do you (not) appreciate/like and why?*”, “*What do you need in order to feel comfortable and safe here in the garden?*”, “*Is there anything that you have found difficult or complicated for you here? And if so, what?*” and “*How have you experienced and used the greenhouse and the terrace?*”

The second interview with the YOD participants was more detailed and in-depth by focusing on aspects concerning (B) Access to nature and surrounding live, using the 19 environmental qualities of QET to guide the interview. The interview included questions such as: “*Is it important to you that there is a lot to see, feel, hear, smell and taste in the garden? And if so, why?*”, “*What do you think about the possibilities for different activities while you are here?*” and “*How do you feel about the importance of secluded places in the garden, where you can be completely alone and don’t have to meet anyone else?*”

The interviews held with the participating staff included questions from both the first and second interview, as described above. During all conducted interviews, with the YOD participants as well as with staff, the questions were kept open and follow-up questions were asked, when necessary, in order to obtain richer and more in-depth material.

### 2.5. Data Analysis

Qualitative Content Analysis was used to analyse the collected data: the interviews with the YOD participants and staff, staff logbooks and diary entries, as well as notes from the staff workshop. This method was used as it aims to gain a condensed as well as a broad description of a phenomenon, where the outcome of the analysis is concepts or categories describing the phenomenon [96]. It is a research method that has come into wide use in health studies [96,97,98]. The result was presented in main categories, categories and new categories. The categorisation within the analysis process followed both a deductive and inductive approach. Since the interviews followed the structure and content of the QET, the starting point of the analysis was deductive [96]. However, each question was left open with follow-up questions when needed to allow for new information, using an inductive approach [96]. The ambition was both to investigate the validity and significance of QET’s environmental qualities in relation to individuals with YOD, i.e., to confirm existing knowledge, but also to develop new knowledge in relation to the target group. The qualitative content analysis was considered well suited for this study as it enabled the synthesis of two seemingly contradictory methodological principles, i.e., openness and theory-driven research, which is appropriate in projects that start from theory and aim to contribute to it [99]. In relation to the current study, it was essential to have an open approach to enable the further development of QET (i.e., our theoretical starting point) and not let the existing version of the tool limit us in relation to the data collection.

Throughout the analysis process, data collected from staff were analysed separately from data collected from the YOD participants. This distinction made it possible to discover potential differences between how the YOD participants expressed their own needs, and how the staff experienced the YOD participants’ needs. All staff data were analysed as one entity during the process, regardless of if it came from spoken interviews, the workshop or in written form from logbooks and diaries.

The analysis was carried out in steps according to Elo & Kyngäs’ [96] three phases: preparation, organisation and reporting. The first author conducted the analysis. To increase credibility, the last author read the analysed material (transcription, coding, categorisation and descriptions). Together the authors discussed the categories and their content until they reached consensus.

#### 2.5.1. Phase 1: Preparation Phase

The recorded interviews were listened to, and the interview transcripts, logbooks, diaries and workshop notes were read through several times in an attempt to get familiar with the data and to get a sense of the whole.

#### 2.5.2. Phase 2: Organization Phase

Data were coded according to the categories. It was then compared against the study’s hypothesis (i.e., that the evidence-based environmental qualities in QET are of importance to individuals with YOD).

All data were reviewed for content and coded according to pre-existing categories: the environmental qualities A1–B13 of the QET.Data that did not fit into the categorisation framework but corresponded to the purpose of the study were assigned new appropriate codes. (This allowed aspects that did not belong to existing categories to give rise to new categories.) After careful consideration, data assigned new codes were either:
Included in existing categories where clear similarities could be found, thereby developing the existing categories (i.e., environmental qualities) and adapting them to the particular target group (individuals with YOD), orAdded as a new category, orAdded to the results as “additional findings”, for example, when related to the well-being and life situation of the YOD participants, creating a broader understanding of the target group. This was used, for example, in the overall description of the main categories: group A. Being comfortable in the outdoor environment or group B. Access to nature and surrounding life.
Tables were developed to structurally organise data by main categories (i.e., group A and B), categories, and new categories. Data from YOD participants and staff were kept separate. Each category now contained descriptions from the collective perspective of all study participants, that is, both YOD participants and staff. However, to understand who said what, a clear distinction was made between the data from the different participant groups.The transcripts were read through again and the coding and categorisations were double checked, to make sure it was done correctly the first time and that nothing had been forgotten, misinterpreted or taken out of its context.

#### 2.5.3. Phase 3: Reporting the Analysing Process and the Results

Category description. A summary was created for each category. The YOD participants and staff perspectives were woven together to describe the quality as comprehensively as possible. To understand who said what, a clear distinction was made between data from YOD participants and data from staff.Order of categories. The order of environmental qualities under the main category group B. Access to nature and surrounding life was discussed and determined by relating the study results to the SOS. The categories represent qualities ranging from most stimulating (B1) to most restorative (B13) for the target group.Specific considerations for the physical environment related to each category were drawn from the category descriptions and placed in separate tables for clarity and easy access. The considerations aim to be useful in practical situations, for example when designing outdoor environments for the specific target group. The format is based on QET and how the environmental qualities are presented there. The QET descriptions were modified, information added or removed, based on data from the current study and the intervention context. A development of QET’s environmental qualities, with a focus on the specific target group, emerged as a result.

### 2.6. Ethical Considerations

Ethical approval for the study was obtained by the Swedish Ethical Review Authority (2016/302). Verbal and written information about the study, its purpose, content and structure were given by nurses at the Memory Clinic to the individuals with YOD who were seen as potential participants, as well as their next of kin. All individuals with YOD were guaranteed confidentiality. They were further informed that participation was voluntary and that they could terminate their participation at any time without it affecting their usual treatment at the Memory Clinic. In addition, opportunities were given to ask supplementary questions to staff at the Memory Clinic who were involved in the project. Signed consent forms were collected prior to the start of the study. Both the written information letter and the consent form was approved by the Swedish Ethical Review Authority.

## 3. Results

The results are presented in three parts:First, an overall description of the two main categories: group A. Being Comfortable in the outdoor environment, and group B. Access to nature and surrounding life.This is followed by descriptions of all 20 categories, within main categories A and B (containing 7 and 13 qualities respectively).Finally, tables summarise specific considerations for the physical environment related to each environmental quality.

### 3.1. Being Comfortable in the Outdoor Environment (A)

This group describes the target group’s clear need to be able to, and dare to, use the outdoor environment, i.e., the opportunity to feel comfortable outdoors, both physically and mentally. The staff explained that this need can be linked to an underlying and ever-present sense of anxiety and uncertainty within the target group. It was what the staff described as a “*…different kind of anxiety and stress…related to a sense of feeling lost*”. The memory and orientation difficulties experienced by the YOD participants, as a result of their dementia diagnosis, may also have influenced this overall need, the results showed. The following categories are included in this group: (A1) Closeness and Easy access, (A2) Entrance and Enclosure, (A3) Safety and Security (A4) Familiarity, (A5) Orientation and Way finding, (A6) Different options in different kinds of weather, and (A7) Calmness.

#### 3.1.1. Closeness and Easy Access (A1)

The YOD participants explained that proximity and accessibility to nature and greenery were considered desirable for them. It was furthermore seen as important to be able to go outdoors easily and independently, even with aids such as a walker. The staff emphasised that easy accessibility to the outdoors, with a lack of physical obstacles, might be of particular importance to individuals with YOD due to their illness, “*…because they have so many other difficulties and [a] lot of, let’s say, locked ‘doors’ and things like that that can be difficult*”. In addition to the lack of physical barriers, staff pointed out that an attractive view from inside a building could influence individuals with YOD to want to go outside.

One individual with YOD described that, in addition to being able to go out by himself with his walker, he also wanted to have access to everything he wanted to see and do in the outdoor environment. The staff explained how this participant had found the accessibility of certain places in the garden challenging, due to his walker, such as when walking through tall grass.

#### 3.1.2. Entrance and Enclosure (A2)

Having a gate for the YOD participants to open and pass through into the garden was important as it made it easier for them to understand where they were “*…because then they know ‘Now we are here’…*” the staff explained (Figure 3). The PLwD appreciated that the garden was enclosed because it made it feel like a safe, protected and cosy place to be in, they said. Staff explained that the opportunity to experience a sense of privacy, away from people outside was important to the PLwD. Clear physical boundaries in the outdoor environment was furthermore seen as a safety issue and an absolute necessity in environments for people with orientation difficulties, by the staff. They also pointed out that the enclosure must be aesthetically pleasing. A painted wooden fence can be perceived as more pleasant than, for example, a metal fence, they said.

#### 3.1.3. Safety and Security (A3)

The results showed that feeling (psychologically) safe and (physically) secure was of particular importance to the individuals with YOD. This was described both by themselves and by staff. The sense of safety emerged as a prerequisite for a positive outdoor experience, as it allowed the YOD participants to spend time in the environment without worrying about being exposed to stressors or harm. Being near other people in the garden felt important, it meant that help was there if needed, for example in case of an accident, and that the fear of getting lost was reduced, the YOD participants explained. Social togetherness and community with people in the same situation were also linked to feelings of safety. In such contexts, the YOD participants explained that they could relax and be themselves. They furthermore described feeling secure in places with good accessibility and good ground surfaces. Environments perceived as calm and beautiful, with vegetation all around, felt safe. Additionally, being in a protected and private place, where the group could be by itself, but without feeling trapped or lacking space, was also linked to safety.

According to the staff, a brain disease such as dementia can affect balance, as well as cause dizziness and drop in blood pressure. Individuals with YOD can furthermore see and interpret things incorrectly, such as misjudge distances in the environment. Therefore, it is important to consider, among other things, ground material and tripping hazards, staff said. Wooden floors outdoors, for example on a terrace, can be slippery when wet and the individuals with YOD may have difficulty judging how slippery it is. Staff explained that smooth and level ground surfaces, as well as normal heights of thresholds and steps, are preferable to slopes, unexpected inclines or gravel that moves a lot under the feet, in order to minimise the risk of tripping. Colours and contrasts on the ground are also important to consider as individuals with YOD sometimes interprets dark fields as holes in the ground, expressed the staff. In addition, it is important for the staff to always have an overview of the whole group. Therefore, it is good to avoid, for example, hedges that are so high that they risk blocking the view.

Water, although much appreciated by the participants, was also brought up as a potential safety risk, by the staff. One individual with YOD stepped straight into the pond to get to the other side, instead of walking around the pond (Figure 4). Another woman jumped back and forth across a stream and got her feet wet, without reacting to it. The staff describe that she did it unconsciously, as if her body moved automatically. “*It’s not like it’s though out by the brain, it’s the body that acts…*”. This constant and automatic bodily movement is common for individuals with YOD, staff explained. With that in mind, there must be opportunities for a lot of movement in the outdoor environment, but without it becoming dangerous for them.

The staff felt that it was particularly important for individuals with YOD to feel safe and secure, as the peace of mind it provides gives their brain a much-needed chance to rest. When there is too much stimuli in the environment they cannot sort through all the impressions, which leads to cognitive fatigue. Therefore, the outdoor environment should not be too messy or cluttered or contain features that may be difficult to read or understand for individuals with YOD.

Already after the first visit to the garden, the staff removed poisonous plants and berries from the site. Several of the YOD participants had quickly and repeatedly put berries and plants, which were often inedible or unripe, into their mouths. This made it clear to the staff that no poisonous berries or plants were to be kept in the garden while the target group was there. A staff member explained that “I don’t think people think about it because all [YOD participants] look so energetic so you say ‘Here you can pick flowers and berries’ but they can’t because sometimes they can’t tell if the red berry should be eaten at all. So you have to consider that if you make a garden [for individuals with YOD] there must [only] be berries that you can eat”.

#### 3.1.4. Familiarity (A4)

An environment that feels familiar can facilitate orientation as it becomes easier to understand. It can also encourage activity and creativity because it is clearer what you can and cannot do there, the staff said. For individuals with YOD, environments that remind them of their childhood (e.g., environments reminiscent of grandma’s house) or other places associated with good memories are preferable “*…it makes you [them] feel at home and safe*” explained the staff.

An individual with YOD pointed out that because they could all relate to [the concept of] a garden, it became a good and easy place for them to meet in. One woman expressed how gardens had always felt like “her place” and therefore she felt at home there. Another saw a connection between her own upbringing, her love of nature and the sense of familiarity she experienced in the garden.

According to the staff, it took longer for the YOD participants to become familiar with the environment than it did for other target groups that had previously stayed in the garden. Not until the last few weeks did some YOD participants feel safe enough to wander off and discover different areas in the garden by themselves. Despite the fact that it took a relatively long time for them to get to know the environment, the staff did not believe that outdoor environments for this target group should be designed as simple as possible and thus be as easy to recognise as possible. Instead, they underlined the importance of offering both safe places that are easy to understand and get to know, as well as exciting places that offer variety and the opportunity to discover and explore. This span in the environment is important, they said. Although, outdoor environments should still look like “*they’ve always looked*”. New, modern and abstract elements can be frightening for individuals with YOD as it leads to difficulties in making sense of their surroundings, the staff explained.

#### 3.1.5. Orientation and Way Finding (A5)

A garden design that minimises the risk of getting lost was pointed out as important by both staff and YOD participants. They expressed that central points in the outdoors were helpful in relation to orientation, such as having various landmarks (for example a clearly visible greenhouse) in the garden. The staff further emphasised the importance of avoiding repetitions in the outdoor environment. Instead, spaces and elements should vary, through different objects, plants and ground surfaces, to facilitate orientation. Places in the environment should provide individuals with YOD with clues about where they are, by looking, feeling or listening to their environment. The design of the environment should furthermore facilitate orientation by, for example, having hedges (not too high to obscure the view) that lead them in the right direction, and paths that bring them back to the starting point. However, too many choices in the environment, such as different routes to take, can make it difficult and confusing for individuals with YOD and should therefore be avoided, the staff said.

The YOD participants experienced impaired orientation skills and had difficulty orienting themselves, especially in new places, they explained. Not recognising themselves or feeling confident that they would be able to find their way back was experienced very stressful and highly confusing. “*… [they] get scared, so there is a risk of ‘No, I don’t go out because I’m afraid I might not find my way home’,…*” explained the staff. One individual with YOD stated that he could only feel a sense of calm and ease in an environment if the “threat” of getting lost was not present. Another individual highlighted the importance of open spaces with full visibility as crucial in her mind to feel safe enough to go outside. Openings and/or gaps in the outdoor environment that allowed for visual contact of central areas, houses and other people also made orientation easier, she explained.

The importance of openness and good overview in relation to orientation was pointed out by staff too. They described how tall hedges grew just outside the entrance to a building in the garden (Figure 5). The group had to pass these hedges as they walked from the building to a greenhouse, which they visited every day in the garden. The staff felt that the hedges blocked the YOD participants’ view, which meant that most did not understand how to get to the greenhouse. “*…it is very, very important… when people [the YOD participants] come out of the entrance to the house, they are not greeted by a garden, they are greeted by hedges… that design is probably not particularly good for this target group … they have stopped there until you walked on with them because they haven’t been able to see, they haven’t had a clue about “Well, where are we going?”*” explained the staff. Most YOD participants only reacted to what they saw at one given moment. Therefore, the environment needed to be easy for them to quickly understand and recognise. They should not have to go around a hedge, for example, to get to a certain place, the staff expressed.

Staff noticed that the YOD participants started to remember more after a while and felt that many of them found it easier to orient themselves towards the end of their stay in the garden. For example, in the last weeks an individual with YOD expressed that “*Yes, but I have been here before, we looked at these flowers last week*”. The staff felt that the YOD participants must be given a chance to land, feel safe, experience repetition and get into a routine, for the possibility of a positive impact on their memory and, by extension, their orientation skills. On one of the very last occasions in the garden, the individual with YOD who had had the hardest time orienting himself was asked to go from one building to another, pick up chili, and bring it back. The man managed to find the right place and bring the chili back. The staff explained that a month earlier this would have never worked, but now he dared.

#### 3.1.6. Different Options in Different Kinds of Weather (A6)

The study took place in autumn and offered the participants a variety of temperatures and weather conditions. The staff expressed that it was important to have outdoor seating that attracted and encouraged the YOD participants to sit down to rest and perhaps get some sunshine on their faces when the sun peeked out. Different forms of covers (such as a terrace roof or parasols) that offered shade when needed and protected the group from rain on wet days were also seen as very useful. During colder days, indoor environments such as a greenhouse were used.

Having access to “transition zones” such as a terrace or a greenhouse is an absolute prerequisite for outdoor interventions with YOD participants because of our climate, the staff explained: “*… [it´s] worth its weight in gold that we can be there [in the greenhouse] because you still get the feeling that you are outside even though you are inside…*” In the garden’s greenhouse, the group was still in touch with nature outside, they could look out into the garden and up into the sky, experiencing both the pouring rain pattering on the windows and the sun’s rays shining in, staff explained. The YOD participants expressed that they appreciated the large number of plants in the greenhouse, and that it felt warm and bright there.

Some of the individuals with YOD pointed out that they were not very sensitive to the weather, and that they wanted to go outside, and did so, even if it was windy and rainy. The staff was of a similar opinion and expressed that the YOD participants did not seem to be very affected by “bad” weather. When it rained, at least half of them wanted to continue walking or working outside, even though they might not have appropriate clothing. According to the staff, this may partly be linked to their love of the outdoors, but also to their potential difficulties in physically dealing with changing temperatures. A delayed reaction to the body’s signals, such as when it is too cold or too hot, can be linked to dementia, the staff explained.

#### 3.1.7. Calmness (A7)

Calm and peaceful environments were perceived as crucial to the well-being of the individuals with YOD, due to their inner anxiety that was easily aroused in situations perceived as hectic or stressful, according to the staff. One individual with YOD said she was happy there was no loud screams or other negative elements such as loud traffic. She described coming to the garden as having the same effect as medicine: “*…it’s just calm here and it’s almost like… a pill… you feel like you should have this all the time instead of pills*”. She laughed and said “*Almost like an advertising slogan. “Come here instead of taking pills!”*”. Another woman explained how the tranquillity and lack of pressure in the garden made her feel at peace in her soul. The sense of calmness and lack of demands was appreciated by several YOD participants, including one man who, towards the end of the stay in the garden, expressed how this had positively affected him: “*I’ve had vibes that it’s going the wrong way for a while, but now I’m back*”.

The inner restlessness of the YOD participants meant that some of them were always on the move, doing things all the time, without having the peace of mind to sit down. The staff saw that they needed active rest, as it benefited their cognitive well-being. Offering places that attracted the YOD participants to sit down was therefore necessary. It was also important to reduce their anxiety by helping them to slow down, focus their thoughts and allow them to be distracted by, for example, a butterfly or rippling water, for increased feelings of well-being. Towards the end of the intervention a calm emerged in the group, which allowed them to work and rest in silence. There was no longer the same need to do something all the time, or to fill in the gaps between activities with something else, staff noticed.

Table 2 summarises specific considerations for the physical environment related to each environmental quality in group A. Being Comfortable in the Outdoor Environment, described above.

### 3.2. Access to Nature and Surrounding Life in the Outdoor Environment (B)

This group represents what makes spending time in the outdoors interesting, enjoyable, rewarding and meaningful. It contains qualities that offer experiences of stimulation, fascination, variety and positive contact with the outdoor environment, including opportunities for tranquillity and restoration. All YOD participants experienced increased loneliness and limited independence as a result of their dementia diagnosis, and by extension reduced opportunities for daily stimulation, social contact and access to outdoor life. The results indicate that this was linked to the very clear need for stimulation in the outdoor environment that emerged during the intervention. This group includes the following categories: (B1) Contact with surrounding life, (B2) Social opportunities, (B3) Joyful and meaningful activities, (B4) Sensory experiences of nature, (B5) Species richness and variety, (B6) Seasons changing in nature, (B7) Culture and connection to past, (B8) Openness, (B9) Symbolism and reflection, (B10) Space, (B11) Serene and peaceful, (B12) Wild nature, (B13) Secluded and protected.

#### 3.2.1. Contact with Surrounding Life (B1)

Although the YOD participants appreciated contact with life outside their private home, for example by being able to sit on their balcony and watch ongoing activities and people walking by, in the garden they expressed a preference for a more sheltered feeling, without contact to the life outside. They enjoyed a more closed off feeling that contributed to a private social community, a sense of “us” and “ours”, the YOD participants explained. The focus was on being in the garden, together with the group. To able to sit and look at the garden itself, enjoy, relax and reflect. Only one of the YOD participants seemed to consider the garden’s surrounding environments and suggested a visit to a nearby park. As a result, the group went for a walk around the park, as well as visited the park’s museum, which was appreciated by all.

The staff felt that having contact with surrounding life and the opportunity to see nature, plants, greenery and animals were of great importance to the YOD participants. It was the close and smaller perspective of their current location (i.e., the garden) that seemed particularly important to them, the staff explained. The YOD participants expressed fascination with life and all living things, such as fish in the pond and unusual trees in the park. However, the staff felt that contact with surrounding life such as shopping streets, schools and traffic would most likely contribute too many different impressions (such as movements and sounds) for them. “*I don’t think …that they can take it in, I think nature is just enough…*” explained one of the staff.

#### 3.2.2. Social Opportunities (B2)

The YOD participants mentioned the importance of having something to gather around. The garden’s environment became a natural topic of conversation for the group. An example of this was the pond where a joint activity could be feeding the fish. Another example was the kitchen garden where the participants, among other things, harvested vegetables together. These became places to meet and talk, the staff explained. They added that going for walks with the YOD participants in and around the garden also had a social function. Even places where the participants could sit and watch the rest of the group, thus maintaining contact and context, offered a social opportunity, according to staff. Activities brought the group together and contributed to social opportunities, the staff explained, even if not all YOD participants always actively participated in the activity itself.

Being part of a group proved extremely important to all YOD participants. A strong connection and sense of community was quickly developed in the garden. The staff explained that meeting others in the same situation, especially other younger individuals who look healthy, was one of the most important things for the YOD participants, who repeatedly said things like “*…here you can just be, I don’t have to think… I don’t have to be afraid of making a fool of myself; I don’t have to worry about what other people will think because I know that everybody knows*”. For that reason, the garden’s most important places became the social meeting places, spacious enough for the whole group, who almost always wanted to sit together, whether just to talk or to participate in joint activities (Figure 6). The YOD participants rarely spent time alone or in small groups of just two, they were often together in a larger group, which made the staff reflect on the need for more places in the garden that could accommodate for the whole group at the same time. Furthermore, the staff described that in case of rain and cold, material from the garden could be taken into the greenhouse where the activities and socialising would continue. However, for one of the women, it felt easier to go for a walk and talk alone with a staff member. Together with the whole group, the woman felt that there was often too much talking going on for her, which could make her feel tired and low.

The group laughed a lot together and often joked with each other. One time an individual with YOD said, “*My head is full, I can’t remember*”, and another commented *“Like Winnie the Pooh—lovely but with a very small brain”*, which led to many laughs. The staff believed it was the individuals’ way of dealing with this difficult disease, and that they needed to do it together with others in the same situation. One individual with YOD expressed his appreciation for all the laughter the group shared and said it was important for them to laugh because “*If you laugh, you cannot cry at the same time*“. The same man made comments such as “*Who would have thought you could have this much fun when you’re sick?*”

#### 3.2.3. Joyful and Meaningful Activities (B3)

Staff highlighted the importance of offering the YOD participants something meaningful to do. It was important for them to feel needed and therefore significant to provide enjoyable and simple but ‘real’ activities that they could manage to do in the garden. The YOD participants themselves expressed how they longed to come to the garden during the weeks and to have something to do. A common experience among them was that their ability to go somewhere by themselves and thus experience stimulation had become limited. This they said was due to their failing memory and increasingly limited independence (for example, they were no longer allowed to drive). One individual with YOD felt that her orientation skills had become worse and therefore refrained from trying new things, such as going for long walks in new places. She missed doing new thing as well as being active and getting her heart rate up. She explained that she felt good when she moved her body and got anxious when she didn’t.

Being active have many benefits for individuals with YOD, according to the staff. The combination of meaningful activities, having something to look forward to, structure in the day and becoming physically tired can help reduce anxiety, depression and the need for medication. The staff explained that, especially as a young person with dementia, it is important to have the opportunity to move around and get physically tired in order to reduce the risk of various behavioural disorders. Having something to do other than just sitting at home is also about having quality of life “*…if you are older … you usually have several diseases and can be tired … but if you are this energetic, you need something else*” explained the staff. Activities with associations to the YOD participants’ childhood, or things done and appreciated by them earlier in life, were valued according to staff. Picking flowers, berries and apples, harvesting and cooking vegetables, sitting around an open fire, feeding animals and replanting plants are examples of activities that stimulated their memories and started conversations. When going for walks, it was important for the YOD participants to have clear, specific and attractive goals, which gave reason for and motivated the walks, the staff shared.

In addition, opportunities for recovery and active rest were also important for the cognitive well-being of the YOD participants, the staff pointed out. Offering a range of voluntary activities in the garden, from social and physical to passive and restorative, was therefore important. Having different activities increased the chance for everyone to find something they wanted to do, but too many options and impressions could make it difficult and confusing for them. For the staff, it was hence important to help the YOD participants to find the right balance so they neither became understimulated nor overstimulated, they explained. According to the staff, the YOD participants did not always have to be involved in the ongoing activity; they could very well be spectators. The activity still brought them together and made it a social and stimulating situation.

At the end of the intervention, it became important for some YOD participants to be able to complete activities, the staff noticed. One man was cutting bamboo sticks into smaller pieces so that they would fit in a bin, but at the end of the day he had not finish all of them. “*Then he comes right in the morning [on the next occasion] and said, ‘I’m picking up where I left off last time’, so that’s actually something big…*” remembered a staff member. The staff noticed that the YOD participants gradually felt safer in the garden, they could get around more easily on their own and they became more active. In the beginning, they mostly stood and watched ongoing activities, they could not go from thought to action on their own but had to be helped to get started. Having difficulty getting started and taking own initiative is associated with having dementia, the staff explained. But the YOD participants’ own “engine” became stronger the longer time they spent in the garden, and they required less and less help, staff experienced. Towards the end, when the YOD participants began to remember activities, they had done earlier in the garden, they could make requests to do them again, and express what they wanted and did not want to do. They also began to take own initiatives in the garden, such as going off on their own to pick flowers. Staff experienced an improvement in their independence, their ability to remember the names of things and how they used to do things in the past. Being in a group and doing activities together provided a sense of connection, context and meaning for the YOD participants, the staff felt.

#### 3.2.4. Sensory Experiences of Nature (B4)

Staff described how the YOD participants showed fascination with many things in the garden, such as the beauty of blooming flowers, the movement of fish in the pond, the size and shape of old trees and new flavours. The YOD participants themselves expressed a desire to experience fascination and stimulation, and to see and do new things. One man described how he felt inspired by a particular tree that had knots on the trunk and branches: “*…that tree … It’s still, I just think it’s inconceivable, I haven’t thought of that before*”. He explained that because he was sick, he wanted to experience as much as possible. Therefore, encountering the special tree had been important to him.

The YOD participants were perceived to be particularly sensitive and receptive by the staff, who noticed that their mood and well-being could easily be affected by positive distractions in the environment. For example, going down to look at the fish swimming in the pond could quickly lift the mood of an individual with YOD when he or she was feeling sad and low, the staff explained. As previously mentioned, many of the YOD participants were quick to taste things like berries and herb-like plants, even though they were unripe or inedible. The staff’s reflection was that being able to taste was something that was important to the YOD participants. They also noted how sensitive they seem to be to taste sensations, due to repeated comments such as “*No, this tastes a little course*” and “*It’s a little oilier*” when for example trying different types of teas in the garden. It was as if they experienced more nuanced differences, as if flavours were more crystallised, staff felt. By tasting what the garden had to offer, the YOD participants were able to understand their surroundings better, the staff believed. Another example of sensory sensitivity was the strong reaction the YOD participants had to traffic noise reaching the group from a nearby highway. The sound was perceived negatively by most of them. Several of the YOD participants repeatedly said they found the traffic noise disturbing, more so than other previous target groups in the garden had, according to staff.

The staff suggested that the sensitivity may be due to the senses taking over when the brain no longer functions as it used to. They observed that despite the YOD participants’ impaired cognitive functions, a strong connection with the senses remained. They therefore felt it was important that the group had opportunities for sensual experiences in nature, such as tasting crops, smelling fragrant herbs or experiencing plants that felt soft to touch (Figure 7 and Figure 8).

It was described by the staff as a form of “garden mindfulness” which could make the YOD participants more aware of the here and now and thus provided mental rest, which was of great importance to them. Sensory stimulation could furthermore trigger memories. An example shared by the staff was the feeling and smell of an open fire, which gave many of the YOD participants a sense of their childhood and led to some of them remembering what it was like to be part of the scouts as children. The staff added that despite the YOD participants’ need and desire for stimulation, it was important to be aware that too much stimulation can be confusing for them, leaving them unable to sort through all the impressions. Too much stimuli can thus have a negative effect on their cognitive functions and lead to mental fatigue.

#### 3.2.5. Species Richness and Variety (B5)

According to the staff, species richness and variety in the environment can be positive for individuals with YOD in many ways. It can provide clues to understanding the season (e.g., trees whose leaves change colour, seasonal crops) and offer a wider range of activities (e.g., harvesting and cooking vegetables, picking apples, feeding animals, picking bouquets of flowers). It can also provide increased opportunities for sensory experiences (e.g., tasting fruit and vegetables, smell flowers, feeling soil between fingers) and be a contributing factor to being enticed to go outside.

One individual with YOD said he found it inspiring to see different species and liked to experience a variety, as it meant there was a lot for him to discover in the garden. He enjoyed seeing life and movements, living things such as trees, animals and water. Staff noted how fascinated the YOD participants were by details in nature, such as plants, flowers, vegetables, animals and insects. Walking around, looking and experiencing the garden was therefore seen as important. Trees seemed particularly interesting the YOD participants, as they repeatedly noticed and talked about different types of trees, the staff explained.

The staff believed that a diversity of species increased the chances for each YOD participants to recognise something, to remember or feel a positive sense of familiarity. One example was a woman who, while picking dill and parsley, remembered her grandfather. She told the story over and over, about her grandfather who [44] was a fisherman, how he smoked buckeye, how she used to angle, and what an idyllic time it was. Another individual with YOD happened to see a hop pergola in the garden and wanted to walk under the hop greenery. The hops were full of cones, and he then remembered how he used to brew beer as a student. “*[It was] cheaper than [it was] good*” he admitted. Staff describe how it was easier for the YOD participants to remember what happened when they were younger than what happened yesterday. Being able to remember provided a sense of safety and capability, which is especially important if you always feel lost, the staff explained.

#### 3.2.6. Seasons Changing in Nature (B6)

Staff explained how the YOD participants had difficulties orienting themselves in time and space, and believed that nature could facilitate that orientation by providing clues “*…you can just look out the window and see when it’s autumn…*” There was a need for the YOD participants to gain an understanding of the outside world, linked to feelings of security, participation and context, as well as quality of life, the staff said. The YOD participants themselves expressed fascination linked to the seasonal changes in nature. One man explained why it was important to them: “*…it becomes part of the whole cycle and nature and …you connect things to something actually happening… something that is continuous…*” He also said that it felt important for him to experience renewal, to have new impressions and to see things happen. New, fun and stimulating things are important to avoid stagnation, he expressed.

Nature’s changes in the garden were experienced through the senses as well as through experiences and activities, according to the staff. Some of the seasonal activities the YOD participants participated in was going for walks, picking autumn leaves, berries and hop cones to make wreaths out of. The participants harvested and tasted seasonal vegetables (e.g., picking and drying apples, harvesting pumpkins and making pumpkin soup) and raked leaves. They noticed how the leaves rustling in the wind made them realise that it was almost autumn. Through the roof of the greenhouse, the group could see birds flying across the sky towards warmer latitudes; a sign that they were headed for a colder season, the group concluded.

#### 3.2.7. Culture and Connection to Past Times (B7)

During the study, certain older or culturally related objects caught the interest of the YOD participants. Things like unusual types of trees, older building features on houses and gates, a garden hammock, and different types of plants and crops planted in and around the garden, according to staff and YOD participants themselves.

Staff furthermore explained that plants tend to help with association and memory stimulation, particularly “traditional” species of e.g., berries, trees or flowers that may have grown in the grandparents’ gardens or in the YOD participants’ own. “*My grandmother always had lots of geraniums*” said one woman, who also started remembering that she used to dig up potatoes in her childhood, when standing in the kitchen garden. When past memories surfaced, the individuals with YOD were always crystal clear when talking about them, said the staff.

At the museum, which the group visited in the nearby park, the group was able to see older agricultural machinery, tools and implements. An individual with YOD whose grandmother had owned a farm started reminiscing about her childhood and remembered various tractors that had been on the farm. One of the men had also “lit up” around the various machines, staff recalled. The same man explained that experiences of culture and connection to past times somehow help to make it all feel meaningful.

#### 3.2.8. Openness (B8)

The YOD participants expressed that open spaces and views in the garden were important for both recreational purposes (the opportunity to look, enjoy, relax and reflect), and for accessibility (being able to see and orientate themselves in the environment) (Figure 9). A pleasant view stops you in your tracks and can contribute with both joy and recognition, said an individual with YOD. They also perceived places that were open as safer. Cramped and confined spaces with poor overview and poor visual contact led to feelings of insecurity, according to one woman.

Staff explained that a pleasant view with interesting features could attract the YOD participants to go outside and make it easier for them to want to have a look around. They added that a good view may provide feelings of being closer to nature, even if, for example, an individual does not have the ability to move much or go very far. Just as the YOD participants, staff pointed out that openness and views were important to individuals with YOD in terms of orientation and understanding their surroundings.

#### 3.2.9. Symbolism and Reflection (B9)

According to the staff, it was likely that the YOD participants were in the middle of a life crisis when they were in the garden, as they had recently been diagnosed with a progressive disease. In a crisis, you need the opportunity for reflection. You are also more receptive to symbolism and metaphors, the staff explained. Some of the YOD participants had personal reflections in the garden connected to their identity and life situation. Staff described how one woman saw herself in a door wreath she was making from rather tough branches. When she was almost done, she surprised everyone by looking at the wreath and stating: *“This is me, sprawling in all directions”*. Another individual with YOD described during an interview that she had noticed a sunflower hanging and looking sad in the garden. She reflected on how it had aged from being yellow and beautiful once upon a time. She felt that maybe the sunflower was there for them, that it kept its spirits up as long as the group was in the garden, to then wither and collapse when the participants were gone. Because the garden was *“… not a garden like the perfect garden but… [an] ordinary garden”*, it was possible to find things that gave rise to personal reflections and symbolism there, she felt. Plants can be good for reflection and for starting conversations, the staff pointed out. It can be easier to talk about the plant than about yourself, they explained.

The staff further believed that symbolism and reflection could function as a form of processing and access to grief, which the individuals with YOD may have been in need of. One woman expressed that she could not quite put her finger on it, but that she felt she had changed during her time in the garden. The staff also noticed a change in her, especially towards the end of the study when she became significantly calmer and a little more serious than in the beginning. She did not joke, laugh and talk as much and started doing things herself in the garden. For example, she carefully and attentively picked bouquets of flowers, completely on her own initiative. The staff also experienced another individual with YOD becoming calmer and more reflective towards the end. She too started walking on her own in the garden, something she had never done before.

#### 3.2.10. Space (B10)

The staff explained that the feeling of entering another world, or a certain space, is a psychological experience. A feeling that comes from stepping into something else, an embracing feeling (Figure 10). This occurred when the participants entered the garden, but also when they sat under a pergola with hanging hops, walked among tall flowers and in amongst an apple orchard, the staff experienced. A sense of space hade also arose as the group sat together around an open fire in a forest-like part of the garden, and as they entered a small, round and igloo-like greenhouse in the garden. The staff pointed out the existence of *“undisturbed worlds within the undisturbed world”*, using the example of the round “igloo” greenhouse. The YOD participants expressed that coming to the garden was like coming to another world “…the whole bubble we live in… it’s a different world…” The environment was a contrast to what they were used to, which had meaning and relevance to the overall experience, they said. Being in a private place that felt like only theirs created a sense of identity and community, with a positive feeling of being part of a “we”, one individual with YOD described it as. The garden felt pleasantly calm and cosy compared to the world outside, *“…you already feel when you put your hand on the handle [of the gate], almost when you enter, that now we are here…”* Coming to “another world” meant the possibility of experiencing something new, according to one man.

#### 3.2.11. Serene and Peaceful (B11)

The YOD participants described how the garden with its plants and rippling water felt serene to them. They said that the water was an important element and that sitting by the pond for a while (preferably in the sun), felt nice and calming. They enjoyed looking at the water, the water lilies and the fish that swam there, which had a particularly calming effect. *“… fish are calm… [then] you feel calm”* explained one individual with YOD. Another woman described the garden as restful due to its beauty, tranquillity and abundance of vegetation. She also linked her experience of the garden as being restful to the fact that gardening had always been important to her.

Although the YOD participants did not seem to have a need to be alone, experiencing tranquillity was significant to them, according to the staff. Down by the pond, for example, the group sat quietly and experienced peaceful moments together.

#### 3.2.12. Wild Nature (B12)

According to the staff, the YOD participants seemed to appreciate that the garden was not ‘too perfect’ and had some elements of a wilder character. Some of the preferred places to walk in was a meadow with tall grass, a shrub forest and groups of large apple trees, for example. The staff pointed out that wild forms of nature also “visited” the garden, such as birds flying in the sky and leaves changing colour in autumn. The staff believed that there was calmness in wild nature, in its undemanding ways, which was positive for the YOD participants, *“…in the wild nature it is allowed to be as it is…”* they explained. At the same time, staff expressed uncertainty about the YOD participants’ experiences of wild nature in the garden and its meaning to them, as they suspected that the YOD participants saw almost all nature as ‘wild’. Staff believed that the preference for wild nature was linked to what the YOD participants had enjoyed earlier in life, before they became ill, and that it therefore depended more on the individual than the target group itself. *“… the environment you grow up in as a child… usually it is the one you actually prefer …that you seek out or that you feel you are comfortable with…”* the staff explained.

One individual with YOD said that it felt important to experience “real nature”. She was fascinated by how nature worked and noticed things when she was out, like how plants grew, how everything came back every year, how ants worked in the anthill and bees flew from flower to flower. According to one man, experiencing nature on its own terms allowed for a variety and provided opportunity for new impressions. Another individual with YOD said she was happy to venture into wilder forms of nature as long as she felt she could walk there, that there were, for example, established paths to walk along.

#### 3.2.13. Secluded and Protected (B13)

Sometimes places that were particularly quiet and still, and at a certain distance from the group, came to be used for rest or for private conversations (Figure 11). But the need for seclusion, that is, to be completely alone, did not exist. Being in contact with the group, even just hearing or seeing the others, was extremely important for the YOD participants’ sense of safety. The staff said that it was only at the very end of the study that some individuals started to explore the garden on their own, by going for a short walk by themselves or just to sit somewhere and look around. Otherwise, they were almost always together. In general, individuals with YOD have a great need for the ability to withdraw, to avoid overstimulation and mental fatigue, according to the staff. But for the YOD participants in this study, all of whom still lived at home and already spent a lot of time alone, socialising while in the garden felt like the most important thing, the staff explained.

Table 3 summarises specific considerations for the physical environment related to each environmental quality in group B. Access to Nature and Surrounding Life in the Outdoor Environment, described above.

## 4. Discussion

This study aimed to investigate and describe the needs of individuals with YOD in a garden setting based on the QET tool [33,69]. The goal was to develop considerations for the physical environment, i.e., relevant knowledge, for the creation of dementia-friendly outdoor environments, while developing the QET and contributing to the field of EBD. In this section of the article, the study will be discussed and reflected on in relation to the Results, Methodological considerations and Future research.

### 4.1. Results

#### 4.1.1. Dementia-Friendly Outdoor Environments

The needs, wishes and preferences of people living with YOD were carefully investigated during the garden study to better understand what a supportive and health-promoting outdoor environment for this target group looks like. It was believed in this study that planning and design of dementia-friendly environments should be based on the experiences of PLwD [44]. People living with YOD were therefore included in the study and among other things, interviewed individually twice to capture their experiences of the garden during the course of the study. The result confirms general descriptions of dementia-friendly environments, such as that of Dementia Australia [45]. However, the current study went beyond listing principles by providing environmental quality descriptions relevant to the specific target group as well as concrete examples in an attempt to bridge the existing gap between research and practice [78]. The results are presented (Table 2 and Table 3) in such a way that designers can use the environmental quality descriptions as a basis for designing supportive and health-promoting outdoor environments for the target group, thereby applying a dementia-friendly approach.

#### 4.1.2. Contribution to Evidence-Based Design

By evaluating and developing the QET into a target group adapted version for people living with YOD, as well as by sharing the results, the current study has contributed to building and developing research evidence, which is an essential part of EBD [66,67]. This new version of QET will in turn need to be tested, evaluated and developed, as well as considered in the context of the specific project where the tool is to be used.

#### 4.1.3. Development of the QET & Environmental Qualities Important to People with YOD

All of the environmental qualities presented in this study have significance for the well-being of individuals with YOD. However, the text below discusses the newly added quality, the qualities that are seen to have a special relevance for the target group, and/or qualities whose descriptions are clearly different from the previous version of QET and thus constitute a large part of the tool’s development. The study confirmed the importance of QET’s environmental qualities for a positive and comfortable experience of being outdoors. However, in relation to individuals with YOD, certain needs emerged that led to the development and creation of a target group adapted version of the QET (Table 2 and Table 3). The pre-existing qualities were primarily developed by adding and/or removing information, according to the study results

##### Qualities in Group A

A new quality emerged, the need for *Calmness*, which was added to the list of environmental qualities. The need for calm permeated all the collected data and was linked to the target group’s inherent and easily evoked anxiety and sense of feeling lost. Since anxiety is such a common symptom in PLwD [100,101], a calm outdoor environment becomes a prerequisite for a positive outdoor experience for the target group. The quality Calmness (A6) was therefore placed in group A, as it should be considered throughout environment. This result is in line with a study by Bengtsson & Carlsson [102] who found Calmness to be a particularly important quality in the outdoor environment for PLwD. Calmness may seem similar to the quality Serene & Peaceful (B11) from group B, but there is a difference between the two. While Calmness needs to be present throughout the environment, Serene & Peaceful is a quality that people seek out, found in some or a few places in the environment (but not everywhere) where it is possible to experience tranquility and restoration. The YOD participants wanted, somewhat unusually, to experience serenity and peacefulness together as a group. The above-described difference between Serene & Peaceful and Calmness has not emerged with this clarity in previous studies.

In relation to Safety and Security (A3), it appeared that experiencing *psychological Safety* (A3a) was very important to the target group. The quality was largely linked to being physically close to the rest of the group in the garden. This proximity meant, for example, a reduced risk of getting lost, which relates to the quality *Orientation and Wayfinding* (A5), and a sense of social safety for them. Being around peers and people who understood them, their illness and what they were going through made them feel safe and relaxed. Overall, staff placed slightly more emphasis on physical security, while the YOD participants emphasised the need for psychological safety to a greater degree. This difference may be due to the fact that the staff, with their experience and knowledge of YOD, had a better understanding of risks and potential consequences in the physical outdoor environment. They naturally also felt responsible for the YOD participants and minimising the risk of anyone being physically harmed in the garden was likely something the staff had in mind.

##### Qualities in Group B

The YOD participants in the garden showed and expressed a clear need for socialisation, *Social opportunities* (B2). This can be explained by the significant reduction in social networks [18] as well as social isolation [17,18,20] that people with YOD often experience [42]. The importance of being part of a group in a non-hospital environment with peers is furthermore confirmed by previous research [42].

*Joyful and meaningful activities* (B3) were also highly valued. At the time of the study, the YOD participants often spent their days alone at home, with feelings of limited independence and missed opportunities for variety and new experiences. These feelings are common to people with YOD [18,20] and may be related to their need for activities in the garden.

The desire to experience stimulation was likewise a prominent finding. The qualities *Sensory experiences of nature* (B4), *Species richness and variety* (B5) and *Seasons changing in nature* (B6), which are experienced as being more restorative for other target groups [74], were experienced as stimulating, fascinating and inspiring by the YOD participants. As a result, these three qualities moved up the gradient of SOS [74] closer to the more stimulating qualities (Figure 12 shows a comparison between the original and developed QET). This emphasis on the more stimulating qualities also meant that the Triangle of supportive environments is turned upside down in relation to the SOS, indicating that the group primarily can be described as sensitive to understimulation [79] (Figure 2).

However, the quality *Secluded and Protected* (B13) in its original description was not sought after in the study. This is because being alone and cut off from the rest of the group was experienced as anxiety-inducing and stressful for the YOD participants. The developed quality description therefore emphasised the need for privacy instead of solitude, as having contact with other participants was a necessity. This constitutes a major difference from other target groups, for example people with stress-related symptoms who often find seclusion and solitude strengthening, while social situations are experienced as exhausting and may produce anxiety [33,103].

Similar to previous research suggesting that memory stimulation can benefit mental and psychosocial well-being for PLwD [54,55], this study found that memory stimulation appeared to increase the ability and sense of self in the individuals with YOD. Activities that had been performed by them earlier in life (such as picking flowers and digging up potatoes) stimulated their memory and led to conversations mainly about the individuals’ childhood and past times. These activities also triggered their “body memory” (the body “remembers” how to do certain activities previously performed, when the individuals were still healthy), which overall seemed to lead to positive feelings linked to independence, ability and self-esteem for the individuals. Possibilities for memory stimulation can be connected to qualities such as *Joyful and Meaningful activities (B3), Sensory experiences of nature (B4), Species richness and variety (B5), Culture and connection to past times (B7) and Symbolism and reflection (B9)*.

The current study results further highlight both the importance of sensory experiences (B4) for individuals with YOD, as well as accounting for a noted sensitivity to stimuli. Great appreciation was shown for, among other things, tasting fruit and crops in the garden, seeing beautiful details in nature and listening to the autumn leaves rustling in the wind. At the same time, distant traffic noise was perceived as disturbing and a sensitivity to stress and too many impressions was exhibited. This may be linked to the high sensitivity to sensory stimulation experienced by PLwD, as a result of impaired cognitive functions [51]. However, while earlier research has identified associations between dementia and certain impairments in relation to taste [51], this study noted that for people with YOD there is a sensitivity associated with taste in the form of heightened taste experiences. The YOD participants seemed to notice *“more nuanced differences, as if tastes were more crystallised”*, as mentioned by staff.

Finally, the purpose of the study was not to evaluate the program, therefore no tests or questions were included to investigate effects on well-being, cognitive function, quality of life, etc. However, an interesting progression was noticed in the YOD participants during the course of the intervention, such as for example, a more developed activity pattern, a higher cognitive capacity, less anxiety and positive emotions, finding which is supported by existing research [40,41]. However, a far as we know this has never been found before in studies particularly focusing on people with YOD. This development had a positive impact on orientation and wayfinding (A5), among other things, and led to a somewhat increased interest in more secluded and private places (B13) in the garden. This progression, which only became evident towards the end of the intervention, prompted reflections and wished from the staff for longer future interventions (suggestions of 12 instead of 8 weeks were put forward), for the opportunity to study possible further progression.

#### 4.1.4. Significance of “Four Zones of Contact with the Outdoors”

The findings of the current study underline the importance of examining the outdoor environment in different zones, in line with the conceptual model of Four Zones of Contact with the Outdoors [33,79]. The results indicate that different zones in the environment provided distinct experiences and had different meanings in relation to contact with the outdoor environment, for the participants with YOD.

An appealing visual connection out to the garden via the kitchen windows, i.e., *zone 1*, turned out to be significant as it enticed the YOD participants to want to go outside. The terrace and the greenhouse in *zone 2* were some of the most well-used places in the garden. They were described as prerequisites for gardening interventions with this target group, especially considering the Swedish weather and that it was easy for the YOD participants to get and/or feel cold. This is in line with the altered experience and behavior response to temperature, commonly experienced by PLwD [49] The warmth that the greenhouse offered, combined with the extensive contact with nature that existed there, was therefore beneficial to the target group. This emphasises the importance of quality *A6. Different options in different kinds of weather*. Outside, the most open part of the garden, i.e., in *zone 3*, was used the most. There it was easy for the YOD participants to get an overview and be able to orient themselves. In this study, *zone 4* consisted of both an inner and an outer part: the inner zone 4 was directly adjacent to the garden (with e.g., apple orchards and flower farms), while the outer zone was a little further away (with e.g., a large park). Environments that offered opportunities to see and do thing outside the garden were perceived as a bonus, as opposed to a necessity. The focus and main interest of the YOD participants was on the garden itself and on being together with the group. However, the directly adjacent inner zone four did enabled longer walks for the group with attractive goals that motivated the walks and stimulated conversations, and the zone was therefore appreciated and visited on repeated occasions.

### 4.2. Methodological Considerations

This study included a relatively small group of people with YOD (six people), where only two different forms of dementia (Alzheimer’s disease and vascular dementia) were present. Although this may raise questions about the generalisability of the results, a smaller group made it possible to carry out an in-depth study, which among other methods used included individual interviews with each staff and YOD participant. The benefit of including both groups of participants is exemplified by findings that emerged during the interviews, where staff placed somewhat greater emphasis on physical safety, while YOD participants more strongly emphasised the need for psychological safety. This is consistent with research showing that both relatives and caregivers often have different opinions and provide different information than PLwD themselves [104]. Furthermore, it relates to the previously mentioned recommended caution against overemphasising safety [58], which risks reducing the highly valued autonomy [9] of people living with YOD. Including both groups opened up more perspectives and reduced the risk of focusing too much on aspects potentially only viewed as important by one of the groups.

With triangulation, additional perspectives and a holistic view in mind, it was important to include staff in this study. The staff had an important role based on their profession, knowledge and experience. The staff could see, follow and get a sense of each individual YOD participants, as well as an overall picture of the group, more so than the participants with YOD could. The staff were also able to put into words things and situations that the participants with YOD may have had difficulty describing and/or remembering. In addition, they could describe things in the light of previous experiences with people with YOD or garden interventions for other target groups.

The size of the study also enabled investigation through method triangulation (observations, interviews, diaries and workshop), data source triangulation (interviews with staff and people living with YOD), as well as a relatively long-term study with the possibility of seeing progression. Although the aim of qualitative research is not to produce statistically generalisable results, methods such as observation and in-depth interviews can be more effective than quantitative methods in exploring complex phenomena and creating understanding of individuals’ experiences, perceptions and behaviours [105], and thereby building evidence for practice [106].

This study primarily used a deductive analysis method, which enables the completion, testing and further development of inductively built models or conceptual systems [97], and therefore has the possibility to advance the research area. This follows the authors’ intention for the QET tool, which was to evaluate and develop the initial version [69]. Using existing theory as a starting point for research can have some inherent limitations and present challenges for researchers, who may be more likely to find evidence that supports rather than does not support the theory [98]. Interview questions were therefore left open and follow-up questions were added when needed to allow for new information, which implies an inductive approach within the overall deductive structure. It was considered important, in the present study, to not start from scratch but to continue the ongoing work regarding health-promoting outdoor environments by using an evidence-based tool, and to contribute to its development.

#### 4.2.1. Site

The rehabilitation garden was perceived to function well for the purpose of the intervention, i.e., to gather knowledge about dementia-friendly outdoor environments for individuals with YOD. This adds to the EBD knowledge base and can be evaluated and further developed in relation to the context of the previously mentioned new roof garden and other future projects.

#### 4.2.2. Recruitment

It is possible that the recruiting nurses may have known or suspected which individuals with YOD had a particular fondness for nature and would like to spend time in the garden. According to Scott et al. [30], recruiting individuals who already have an interest in gardening and outdoor living could potentially influence study outcomes that focus on gardening. Previous research also indicates that a continuous interaction with nature early in life increases the ability to develop attachment and consideration for animals, nature and the environment [107]. However, since the current study does not focus on the effects of outdoor activities in relation to the physical and psychological well-being of PLwD, but instead on needs and use in relation to the outdoor environment, we consider the risk of misleading results to be relatively small.

Conducting research involving PLwD poses challenges and raises ethical questions. With this in mind, precautions were taken, and consideration was given to the research group’s ethical awareness and preparedness in relation to this target group, for example in relation to the recruitment process as well as applying for and obtaining ethical approval.

#### 4.2.3. The Interview Situation

Some challenges were experienced in connection with the interviews conducted with the individuals with YOD. Although they were all very positive to being interviewed *(“We create everyday history”* as one individual with YOD said), it was sometimes noticed that the interviewee began to lose focus and energy as the interview progressed, and that they occasionally had difficulty remembering what the garden looked like or what activities they had participated in, during the interview itself. Attention and concentration lapses, fatigue and memory loss are related to dementia, and can affect an interview situation [106]. Despite these somewhat complicating factors, solid data collection from (and about) the YOD participants was achieved. The interviews and workshop with the staff, as well as their notes (where the YOD participants’ comments and feelings in the moment were captured) worked well as a complement and gave us a comprehensive picture of the participants’ time in the garden, as also described by Svanström & Sundler [108] and Samsi & Manthorpe [106]. The notes also aided the analysis to some extent by contextualising the interview responses. The combination of methods thus worked as intended.

### 4.3. Future Research

This intervention involved a relatively small group of people with YOD and few dementia diagnoses, allowing for a detailed and in-depth research study. However, we believe that more studies of this kind should be carried out, for the possibility of gaining more knowledge about this particular target group in relation to the physical outdoor environment, and to increase generalisability. These studies should preferably extend over longer periods of time (for example 12 weeks instead of 8, as suggested by the participating staff) and could include a greater variety of dementia diagnoses, or alternatively focus on other diagnoses, other stages of the illness and/or other outdoor environments, than in this study.

Exploring the increased sensory sensitivity noted in the YOD participants may also be an interesting focus for future studies, to further specify the target groups’ needs in relation to sensory stimulation in the outdoor environment. This may be useful for practitioners and future design projects. In addition, it may be interesting to study individuals with YOD raised in urban settings without any specific memories of nature to find out how they would experience and react to spending time in a natural environment.

## 5. Conclusions

This study has highlighted specific considerations for the physical environment of individuals with YOD, for the possibility of creating supportive and health-promoting, i.e., ‘dementia-friendly’, outdoor environments. This was done through the development of QET, into a target group adapted version of the tool, for use in future design projects. 20 environmental qualities of importance for people with YOD was presented in two groups: group A, containing 7 qualities related to being *Comfortable in the outdoor environment*, and group B, containing 13 qualities concerning *Access to nature and surrounding life*. The B-qualities are placed on a gradient, from most stimulating (B1) to most restorative (B13). The main findings of this study are concluded below.

An environmental quality was added to the QET, *Calmness* (group A), which was perceived both fundamental and crucial to the well-being of individuals with YODAs the individuals with YOD were quick to taste what the garden had to offer including things that could be unripe or even inedible, it is important to avoid poisonous berries or plants in outdoor environments for the specific target group.The more stimulating qualities emerged as the most desired and appreciated ones by the individuals with YOD, turning the Triangle of supportive environments model upside down in relation to the Sensory Opportunity Spectrum (group B), indicating that the target group can be described as sensitive to understimulation.A certain need for more restorative qualities was experienced towards the end of the intervention, and a sensitivity to certain stimulation was also noted in the study.There seem to be no need or desire for seclusion (i.e., solitude) for the participants with YOD. Instead, they wanted to experience peace and tranquility together with the group. This contrasts with previous versions of the QET and other target groups in the garden.The study confirms the importance of including both individuals with YOD and experienced staff in research studies of this nature, for a broader perspective and increased opportunity for a comprehensive understanding.

## Figures and Tables

**Figure 1 ijerph-21-01047-f001:**
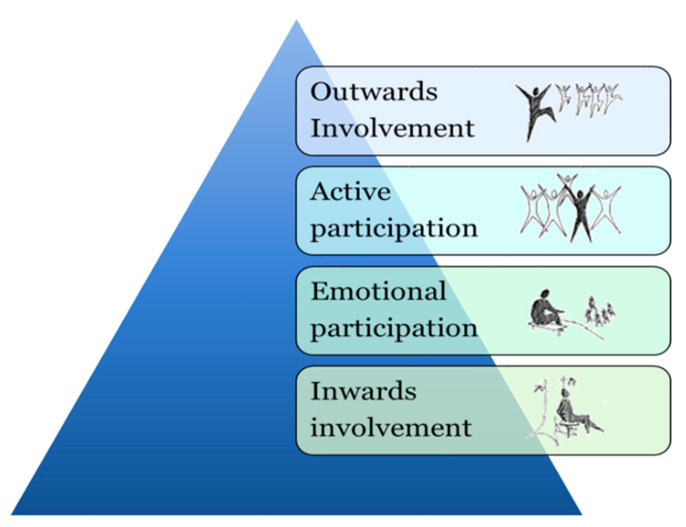
Adaptation of Triangle of Supportive Environments [74].

**Figure 2 ijerph-21-01047-f002:**
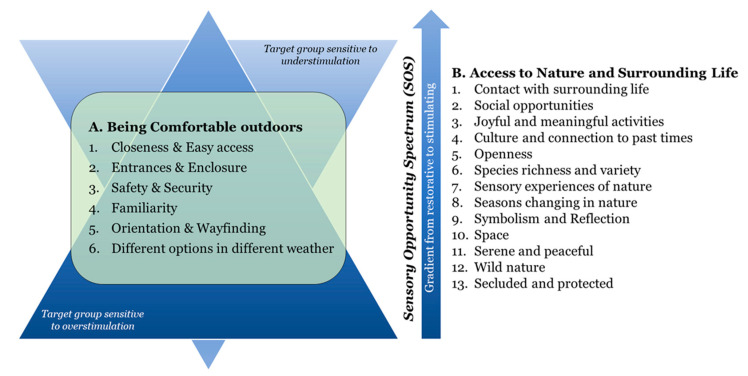
Adaptation of the 19 environmental qualities from Quality Evaluation Tool (QET) integrated into the Triangle of supportive environment model [33]. The upside-down triangle illustrates that the more stimulating qualities are the most desirable for target groups sensitive to understimulation.

**Figure 3 ijerph-21-01047-f003:**
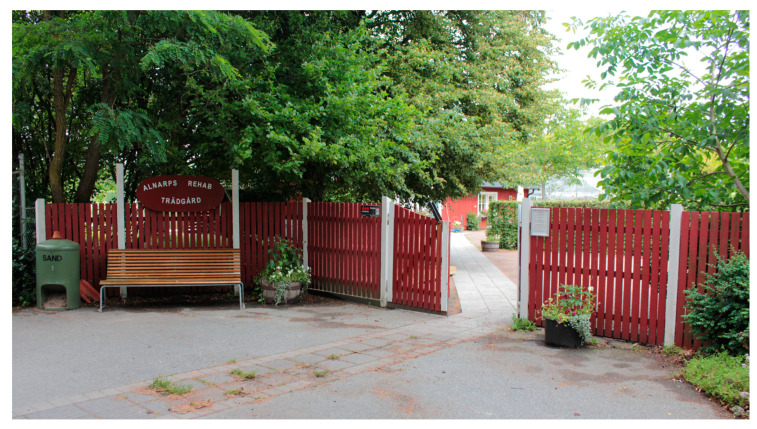
For individuals with YOD, a gate to the garden that must be opened and walked through can provide clarity about one’s whereabouts. Reproduced with permission from Anna Åshage.

**Figure 4 ijerph-21-01047-f004:**
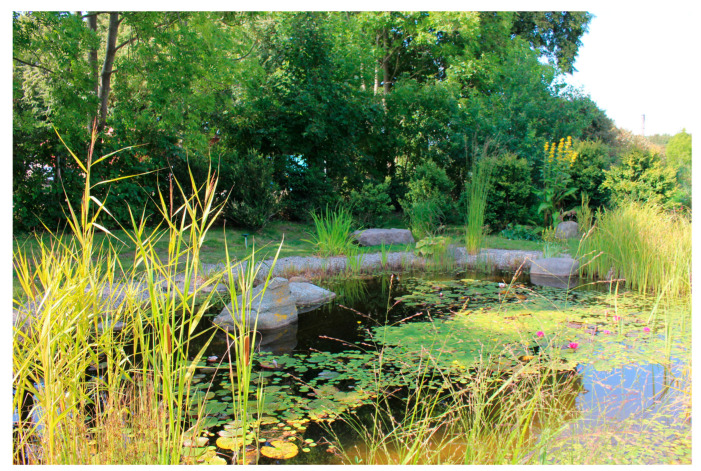
The garden pond. Reproduced with permission from Anna Åshage.

**Figure 5 ijerph-21-01047-f005:**
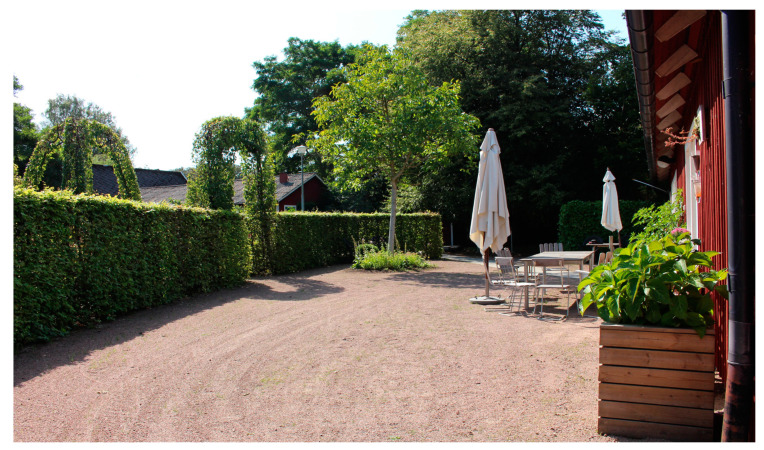
High hedges that were perceived to block the YOD participants’ view, making it difficult for them to orient themselves. Reproduced with permission from Anna Åshage.

**Figure 6 ijerph-21-01047-f006:**
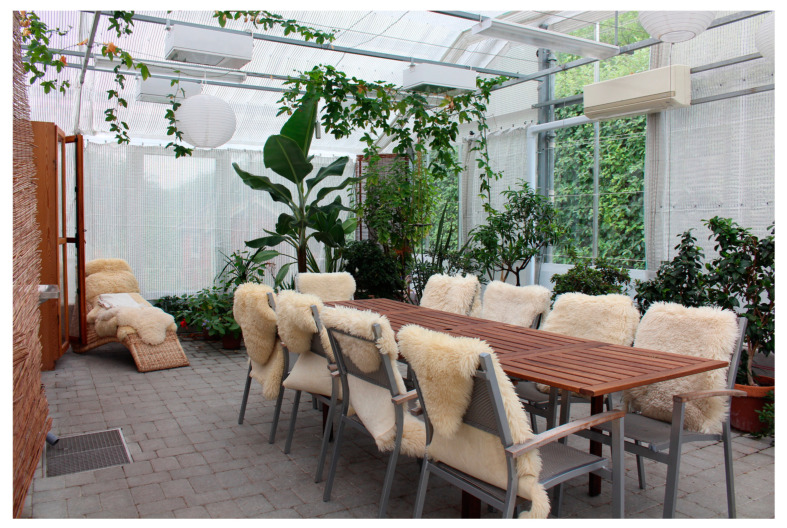
Social opportunities: seats for the whole group to gather and sit down together in the greenhouse. Reproduced with permission from Anna Åshage.

**Figure 7 ijerph-21-01047-f007:**
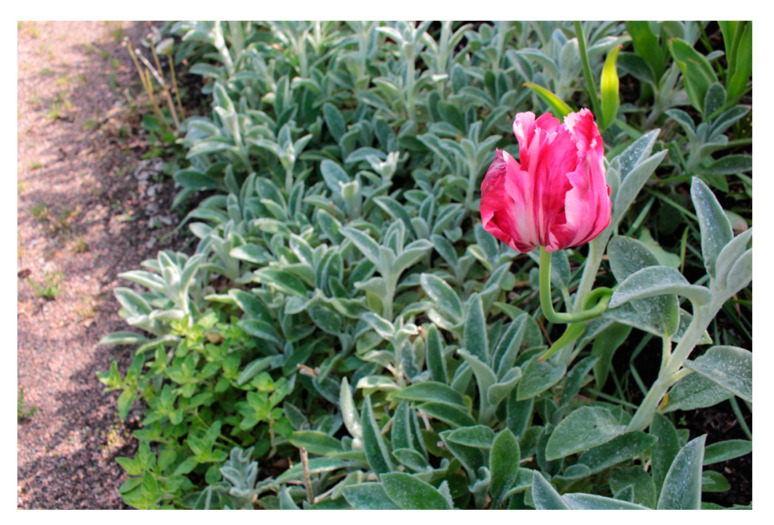
Sensory experiences in nature: colourful flowers and leaves that feel soft to touch. Reproduced with permission from Anna Åshage.

**Figure 8 ijerph-21-01047-f008:**
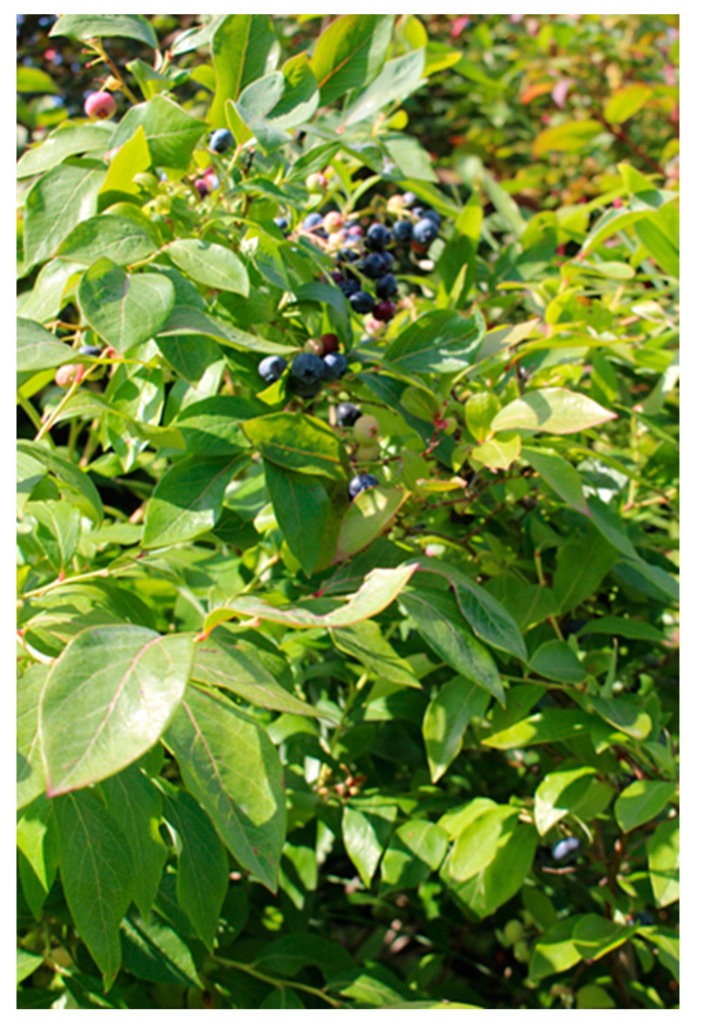
Edible berries growing in the garden offer different taste sensation. Reproduced with permission from Anna Åshage.

**Figure 9 ijerph-21-01047-f009:**
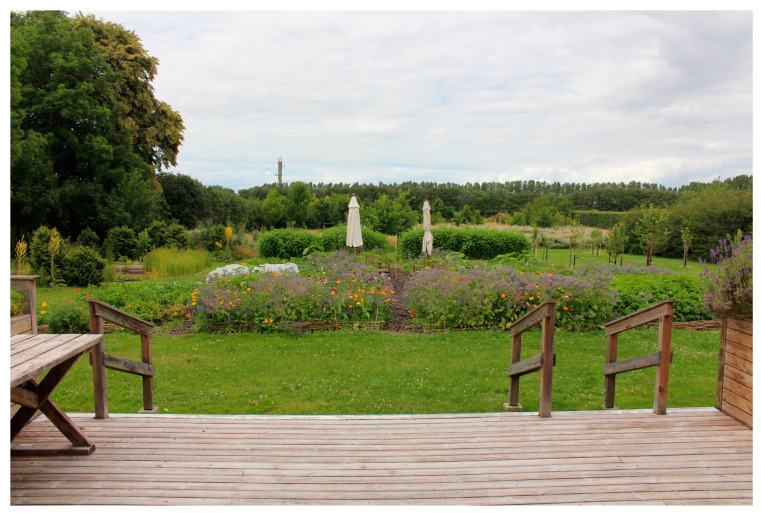
Open spaces and views in the garden were important to the YOD participants, both for recreational purposes, as well as for accessibility and orientation. Reproduced with permission from Anna Åshage.

**Figure 10 ijerph-21-01047-f010:**
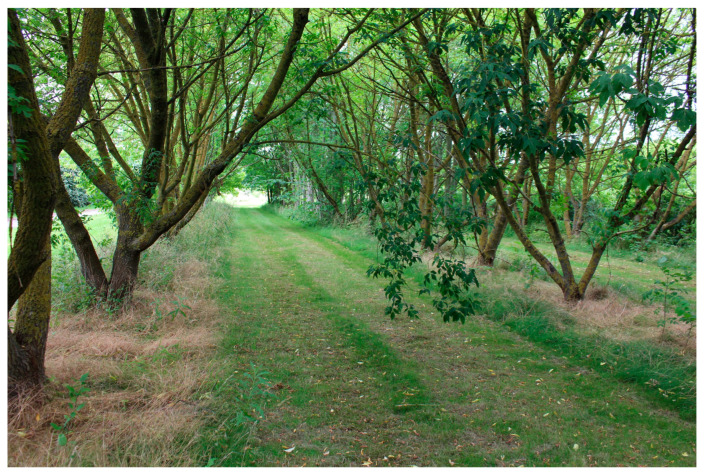
A sense of Space. Reproduced with permission from Anna Åshage.

**Figure 11 ijerph-21-01047-f011:**
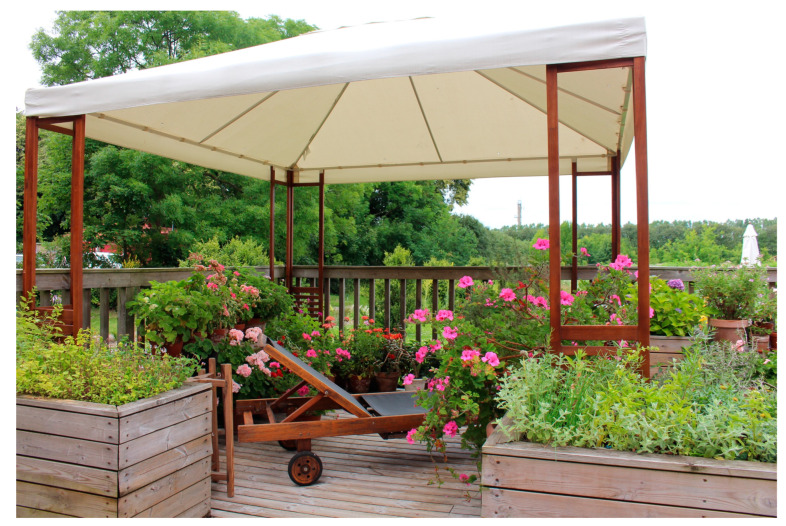
A quiet place some distance from the group, but where the others can still be seen and/or heard, can feel like a good and safe place to rest for individuals with YOD. Reproduced with permission from Anna Åshage.

**Figure 12 ijerph-21-01047-f012:**
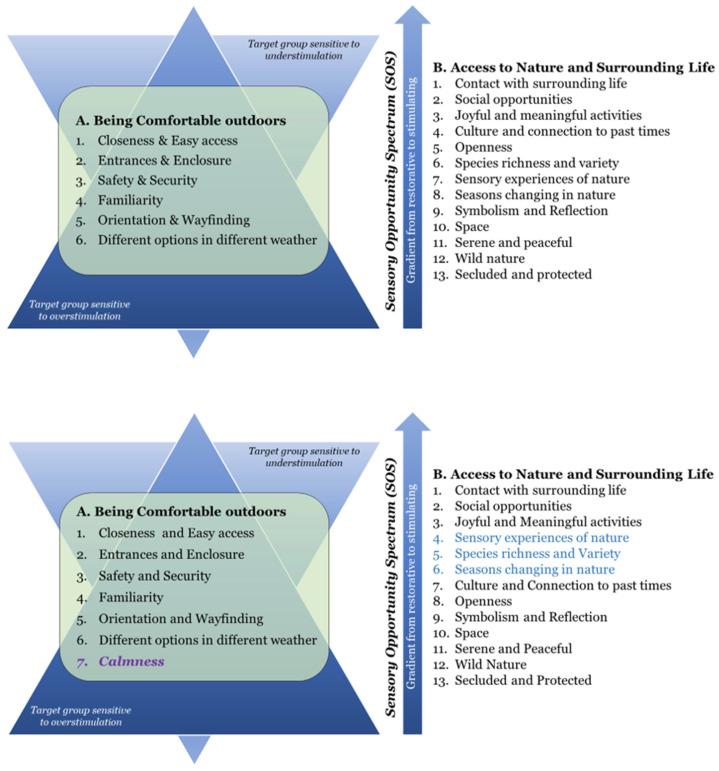
Comparison between QET [33] (**top** figure), and the current study’s target group adapted version (**bottom** figure), where the additional quality *Calmness* (group A) is highlighted in purple and the three qualities that have moved up the gradient (group B) are highlighted in blue.

**Table 1 ijerph-21-01047-t001:** Demographic characteristics of the individuals with Young- Onset Dementia (YOD).

Characteristics	Individuals with YOD
Age (years)	
- Mean - Range	61.5 51–67
Gender, n (%)	
- Women - Men	4 (67%) 2 (33%)
Type of dementia, n (%)	
- Alzheimer’s disease - Vascular	5 (83%) 1 (17%)
Severity of dementia, n (%)	
- Mild cognitive decline - Mild- moderate cognitive decline - Moderate cognitive decline	3 (50%) 1 (17%) 2 (33%)
Use of walking aids, n (%)	
- Walker	1 (17%)

**Table 2 ijerph-21-01047-t002:** Group A. Being Comfortable in the outdoor environment.

A. Being COMFORTABLE in the Outdoor Environment Seven Environmental Qualities, Which Constitute Considerations of the Physical Environment, for Designers to Pursue in Therapeutic Gardens. These Qualities Have Characteristics That Support People with YOD’s Ability to Dare to Go Outside, and Feel Comfortable Enough to Use the Environment. They Are Important to Consider Throughout the Green Area so That Everyone Can Use the Space on Equal Terms.
A1. Closeness and Easy access There is a nearby green outdoor environment (e.g., garden) for the target group. It is clearly visible through windows and easily accessible from the building where the target group is staying. It is easy to get in and out independently, even with aids such as a walker, taking into account doors, locks, thresholds etc.
A2. Entrance and Enclosure The enclosure of the outdoor environment (hedges, fences, etc.) corresponds to the level of security and safety that the target group needs without being perceived as restrictive or confining. There is a clearly visible and welcoming entrance, e.g., a gate that must be opened and passed through, making it easier for target groups with orientation difficulties to understand where they are.
A3. Safety and Security (a) Safety: the risks of *psychological discomfort* in the outdoor environment are very small. The outdoor environment is appealing, without intrusive impressions such as loud noises or a lot of movement that can be interpreted negatively by the target group. Consider risks of outsiders intruding and risks of being viewed against one’s will by outsiders.
(b) Security: the risks of *physical discomfort* in the outdoor environment are very small, such as the risk of tripping, slipping or falling into water. There are no poisonous plants and berries. Ground coverings are available in terms of width, edges, slopes, surfaces and materials. Colouring and contrasts are of such a nature that the target group does not misinterpret e.g., dark fields to be holes in the ground. Steps and thresholds are of normal height.
A4. Familiarity The green area feels like a natural part of the setting. It is easy to see and get to know the outdoor environment. Its character, content and usage possibilities are familiar and easy to understand. Elements that are new, modern and abstract are avoided, as they can be perceived as intimidating and difficult for the target group to grasp.
A5. Orientation and Wayfinding Design and layout of paths, places and landmarks are clear and help people understand and orient themselves in the outdoor environment. It is important that hedges (that are not too high and obscure the view), paths and walkways lead back to the starting point and not to dead ends. There is a good overview that makes it easy to read and understand the environment. The outdoor is varied and there is a range of features, such as objects, plants and ground surfaces. The variation in the environment provides important clues that can facilitate orientation.
A6. Different options in different kinds of weather Footpaths and seating areas are positioned so that there is possibility of sun, shade and shelter from wind and rain. Flexible chairs that can be moved around are available. In case of rain or cold weather, transition zones such as a greenhouse offers weather protection, as well as vegetation, heat, light and good contact with the outdoor environment.
A7. Calmness The environment is calm, peaceful and undemanding. It is neither overcrowded nor has unexpected or disturbing features, such as heavy traffic or loud voices. It is clear who uses and has access to the environment.

**Table 3 ijerph-21-01047-t003:** Group B. Access to nature and surrounding life in the outdoor environment.

B. Access to Nature and Surrounding Life Thirteen Environmental Qualities, Which Constitute Considerations of the Physical Environment, for Designers to Pursue in Order to Increase the Possibility of Both Stimulation and Restoration for People with YOD in Therapeutic Gardens. These Qualities Follow a Gradient in the Sensory Opportunity Spectrum (SOS) of Supportive and Health-Promoting Environments—from Most Stimulating (B1) to Most Restorative (B13).
B1. Contact with surrounding life The environment provides rich opportunities to focus on the small and close perspective of the natural environment, such as plants and animals. The focus is on the current location. There are opportunities to visit the environments and/or society outside, to take part in the life and experience the activities that take place there.
B2. Social opportunities There are opportunities in the outdoor environment for activities and places where you can meet, talk or just watch other people. In these places there are, for example, plants, water features, distinctive environments or elements to gather around, look at and talk about. There are seating options that make it easy and pleasant to meet and socialise outdoors. There are places where the whole group can gather and sit together, but also places for just a few to meet. Paths allow for walks where two or more people can walk together.
B3. Joyful and Meaningful activities There are places in the outdoor environment for sedentary activities (such as relaxing or resting, reading, drinking coffee, reflecting, watching and listening to nature and other people), social activities, physical activities, therapeutic activities and gardening activities. These places are large enough for a group of people to be active at the same time. There are walking paths, preferably with a clear goal, which can be used for exercise as well as for leisurely walks. The activities offered correspond to the wishes, needs and abilities of the target group.
B4. Sensory experiences of nature There are opportunities to get close, experience and be fascinated by nature with all one’s senses, by seeing, feeling, hearing, smelling, tasting and enjoying what nature has to offer. Trees, plants, flowers, fruits, berries, herbs, animals and insects are some examples. Experiencing natural elements such as sun, sky, wind and water is possible. There are areas that offer alternatives in stimulus intensity, regarding e.g., colours, sounds and scents, to avoid both over- and under-stimulation for the target group.
B5. Species richness and Variety There are areas with species richness in terms of plants and/or animals that give varying expressions of life. There are opportunities to be fascinated by details in nature and to see, learn and experience new things. Species richness and variety in the outdoor environment provide a wide range of activities, an increased opportunity for sensory experiences and memory stimulation, and contribute with important clues about the season for the target group.
B6. Seasons changing in nature There is an opportunity to follow the year’s changes in nature, partly with your senses but also through experiences and activities in the outdoor environment. During autumn, examples of this can be raking leaves, harvesting, cooking and tasting what is available (e.g., pumpkins or autumn apples), sitting around an open fire, or tying bunches of flowers and autumn leaves. This gives clues to people who have difficulty orienting themselves in time and space.
B7. Culture and Connection to past times There are places in the outdoor environment that provide an opportunity to be fascinated by human history and culture. There are objects that stimulate the memory, such as cultivation lots, a barbecue area, old tools, machinery and building details. Berries, flowers and crops can also contribute with associations to previous experiences (*“my grandmother used to grow these”*).
B8. Openness There are inviting open green areas with views of nature and plants. The view is also accessible from inside (e.g., via window) and has the potential to entice people to want to go outside. There are open areas overlooking central parts of the outdoor environment that facilitate accessibility and orientation for the target group.
B9. Symbolism and Reflection There are elements in the outdoor environment that can give rise to symbolism and metaphors between one’s own life situation and nature. Reflections evoked by a sunflower that perseveres and remains brave even though it is no longer yellow and beautiful is an example.
B10. Space There are areas that give the feeling of entering an undisturbed world or coherent whole. A private garden, an apple orchard or a peaceful greenhouse are some examples. These areas allow the individual or group to be by themselves and to feel that the place is only theirs.
B11. Serene and Peaceful There are peaceful and well-maintained places in the outdoor environment with soothing elements of water and/or greenery that offer relaxation. Seeing water with associated plants and animal life, such as sitting by a pond and watching fish swimming is an example. The sound of rippling water is especially soothing. It is possible to experience tranquility and be close to other people at the same time.
B12. Wild Nature There are opportunities to experience nature on its own undemanding terms. There are areas where plants appear to have come by themselves and where they have been allowed to develop freely, e.g., a meadow with tall grass or large apple trees. There is untouched wildlife as part of the natural environment, such as birds and insects.
B13. Secluded and Protected There are surrounded and protected green places where you can do what you want undisturbed, be alone, have private conversations, rest or just watch other people from a distance. Visual contact with central parts of the environment, as well as seeing and/or hearing other people even from more secluded places is always possible. There is comfortable furniture that allows for rest.

## Data Availability

The data presented in this study are only available for the purposes of peer review due to privacy and ethical reasons. Requests to access the data should be directed to the corresponding author.

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
