# Peer review of "The Design of Health Promoting Outdoor Environments for People with Young-Onset Dementia—A Study from a Rehabilitation Garden"

_ijerph, 2024, doi:10.3390/ijerph21081047_

Round 1

Reviewer 1 Report

Comments and Suggestions for Authors

Dear Authors

Thank you for giving me the opportunity to read and review this paper entitled “The Design of Health Promoting Outdoor Environments for 2 people with Young-Onset Dementia - A Study from a Rehabilitation Garden”. It is an interesting and relevant study/paper.

However, there are some issues and inadequacies in the paper that I believe must be addressed to develop the paper for publication. It is a comprehensive and lengthy paper, so be aware of the balance. In some parts you need to prioritise and be more precise.

I have organised my comments in relation to the structure of the paper: introduction, materials and methods, results, discussion, and conclusion.

Introduction

The presentation of dementia focuses on the amount of people with dementia and the prognoses. It is relevant to understand the prevalence of dementia. However, this alone is insufficient when aiming at studying and understanding how the environment can be supportive for Young-Onset Dementia (YOD). Therefore, the paper needs to explain the disease especially in relation to sensory stimulation and memory. In line 1329-1340 you write about it, but it should be frontloaded.

An understanding of the qualities of the physical environment is missing. In paragraph 1.4 the characteristics of the dementia-friendly environments should have been described to establish an overview of existing literature, especially since this paper focuses this theme. See also the comment in relation to the discussion.

The aim of the paper/the study is to discover knowledge about the needs of people with YOD that can be used in future designs based on the approach of Evidence-Based Design (EBD). The chosen approach and method to the study is the Quality Evaluation Tool (QET). The choice of QET should have been substantiated. Why is this approach relevant in relation to YOD patients?

However, it does not make sense to use nearly a whole page on an introduction EBD (1.5. Evidence-Based Design) because this study is not specially about EBD – it a premise. This paragraph should be shorter and more focused. Furthermore, the approach to the study using QET should be substantiated instead, particularly in relation to scientific research, because currently the paper focuses very much on practitioners. The weighting should be opposite. Furthermore, the paper needs to answer what this approach can do different from others e.g. a more explorative approach.  

Line 202 -204: Three different theories which underlie the Quality Evaluation Tool (QET) are mentioned without any introduction. You could introduce these theories to substantiate the choice of QET.

Figure 1 and Figure 2 are used to introduce QET and provide an overview of the content in the model. This introduction is quite implicit and superficial, considering the crucial role QET plays in the study. The different concepts should have been described or elaborated upon, as done in reference number 34 where table 3 and 4 present the 19 qualities. In line 1280, you write that you have adjusted (“developed by adding and/or removing information”) the exiting QET qualities. Such an approach implies a description of the material being adjusted.

Reference 34 distinguishes between low stimulation qualities and high stimulation qualities in relation to the 13 qualities in category B and has drawn a stippled line in the model (even though it does not correspond to table 4). Why have you not mentioned or commented on this distinction? I will return to this list of qualities.

It surprises me that the hypothesis (mentioned in line 517) and QET are not integrated in paragraph about the aim since QET plays a crucial role in the study. Operating with a hypothesis should be frontloaded.

Materials and methods

The theoretical basis for content analysis should be justified.

In line 311, the authors writes that the typical approach of qualitative research incorporates randomization or control groups. I can see that this statement is supported by a reference. However, there exist other fields of qualitative research, particularly outside the medical world, such as in social sciences, that operate without control groups.  

Line 397. It seems unnecessary to operate with the category of participants, when you only use it very few times.  

The use of semi-structured qualitative interview as a method should be substantiated to show the foundation you are building on.

A general theme is the use of the staff as a decisive source. The paper needs to describe why this approach was chosen and to reflect on this choice and its consequences, especially considering a research design where one of the authors could have participated as a staff member (doing participant observation) to gain a direct impression of the YOD participants and their interaction with the garden.

Did you find differences between the YOD participants or the staff? (you mentioned this possibility in line 495)

Results

It is a deficiency that figure 1 and Figure 2 are not used graphically in the presentation of the result or in the discussion to emphasise the contribution of this paper.  You could have clearly shown the position of the qualities that differs from the QET model (line 1320) in a clear way. I would recommend using figures more actively.

Discussion

I find the discussion a bit too automatic. Meta text is needed to introduce the structure of the discussion, thereby indicating a conscious decision about what to discuss.

Some of the qualities mentioned could have been summed up in the result section emphasise the contribution, such as the link between Openness and Orientation (line 1302-1305).

In line 1259, you present a description of dementia-friendly environments. I understand that you want to use it to emphasise that your result confirms exiting knowledge. However, the presented description is so general that it does not make sense here. You could have used this description as an introduction to paragraph ‘1.4 Dementia-friendly environments’. This paragraph that is quite brief compared to some other parts of the papers and should be developed because of the aim of the study.

A prominent finding and contribution to the QET model is the need for Calmness as an extra quality. A quality that reference 88 has also identified in relation to people with dementia. However, a reflection on whether this need is specific to this group or if the QET has failed to recognize this quality is missing.

The introduction argued that we need knowledge about the need of people with YOD because the literature focus on older people with dementia. The discussion should have provided clearer insight into the difference between the results of this study compared to other studies. The study has shown that the YOD participants are very interest in the community around the group (the social part) which could have been discussed here. Additionally, the study indicates that YOD participants have a special sensory sensitivity compared to other groups (line 1007). It could have been interesting to reflect on this in relation to a future design and how practitioners can correspond to that.

The discussion needs a reflection about how knowledge from a two-hectare garden on the ground can be used in the design of a roof garden.

The childhood plays a role for the YOD participants. It could have been interesting to study YOD participant raised in the city without any specific memories of nature to find out how they reacted to nature. I miss a reflection about that. Furthermore, it could have been interesting to reflect on the outcome if the study has lasted more than eight weeks.

Conclusion.

The conclusion is very general. Unfortunately, it does mention that the study is based on the QET model and does not emphasise the additional category, which I find is an important contribution.

Author Response

Dear Reviewer, 

You can find our reply in the attached document. 

Kind regards

Nina Oher

Reviewer 2 Report

Comments and Suggestions for Authors

A very detailed study looking at a subgroup of dementia patients not previously assessed in this way. Reported according to recommended guidelines. Does contribute to body of knowledge given constraints of the study. Limitations adequately addressed. Well referenced. I think this manuscript adequately achieved its goal to explore and share garden design recommendations for people coping with YOD. Small 'n' limits generalizability. A great start.

Although it is a qualitative study, it does run a little long. There is some repetition of wording and phrasing, which can be slightly reduced. 

103 add 's' to intervention

107 I think the 'outdoors' is a singular entity and 'has' the potential.......

The paper is well written. The methodology is well described and triangulation helped increase richness of data. The use of interviews with caretakers offers convergent validity which is important considering the cognitive abilities of the participants and low number of subjects. Results are consistent with previous related research with dementia population. Recommendations seem appropriately derived from data and reflect identified needs of the population. Limitations addressed properly. The manuscript is long which is typical for qualitative studies. I do believe it can be condensed/reduced here and there if length is of issue.

Section 1 can be reduced here and there with less detail presented. First and last sentences in a number of paragraphs while contributing to a smooth narrative could be eliminated without affecting quality of info presented.

Sect. 1.7 first sentence could be eliminated start with... needs being underexplored

2.1 last sentence can be eliminated

2.4.1 mentions conducting 2 interviews sentences can be combined here

2.5.1eliminate first sent.

Author Response

Dear Reviewer, 

Please find our reply in the attached document. 

Kind regards

Nina Oher

Reviewer 3 Report

Comments and Suggestions for Authors

In the manuscript titled The Design of Health Promoting Outdoor Environments for people with Young-Onset Dementia - A Study from a Rehabilitation Garden, through a qualitative method to investigating outdoor situations that are helpful to the health of persons with dementia, to promote research in the field of "dementia-friendly" outdoor environments.

The paper offers useful information based on current events, however it falls short in explaining the significance of its concepts. The article's structure is quite steady, however for the study issue to be understood more clearly, the language must be coherent and the reasoning must be organized. The study questions' results were poorly presented throughout the discussion, and summative evaluation was absent despite a summary of the conversation on the pertinent subjects. To improve the experiment's use in the future, the authors could offer more detailed instructions for applying the methodology.

It's possible that the number of individuals recruited won't be enough to adequately represent the study population as a whole.

Authors must revise their writing in response to recommendations.

Author Response

(The authors gave the same response as above.)

Round 2

Reviewer 1 Report

Comments and Suggestions for Authors

Thank you for the revisions. The paper has been improved.